EMBO
Molecular Medicine

# DNA methylation inhibitor attenuates polyglutamine-induced neurodegeneration by regulating Hes5

Naohide Kondo[1,†], Genki Tohnai[1,†], Kentaro Sahashi[1], Madoka Iida[1], Mayumi Kataoka[1], Hideaki Nakatsuji[1], Yutaka Tsutsumi[1], Atsushi Hashizume[1], Hiroaki Adachi[2] (iD), Haruki Koike[1], Keiko Shinjo[3], Yutaka Kondo[3] (iD), Gen Sobue[4,*] (iD) & Masahisa Katsuno[1,**] (iD)

## Abstract

Spinal and bulbar muscular atrophy (SBMA) is a polyglutamine-mediated neuromuscular disease caused by a CAG repeat expansion in the *androgen receptor* (*AR*) gene. While transcriptional dysregulation is known to play a critical role in the pathogenesis of SBMA, the underlying molecular pathomechanisms remain unclear. DNA methylation is a fundamental epigenetic modification that silences the transcription of various genes that have a CpG-rich promoter. Here, we showed that DNA methyltransferase 1 (Dnmt1) is highly expressed in the spinal motor neurons of an SBMA mouse model and in patients with SBMA. Both genetic *Dnmt1* depletion and treatment with RG108, a DNA methylation inhibitor, ameliorated the viability of SBMA model cells. Furthermore, a continuous intracerebroventricular injection of RG108 mitigated the phenotype of SBMA mice. DNA methylation array analysis identified *hairy and enhancer of split 5* (*Hes5*) as having a CpG island with hyper-methylation in the promoter region, and the Hes5 expression was strongly silenced in SBMA. Moreover, Hes5 over-expression rescued the SBMA cells possibly by inducing Smad2 phosphorylation. Our findings suggest DNA hyper-methylation underlies the neurodegeneration in SBMA.

**Keywords** DNA methylation; epigenetics; Hes5; RG108; spinal and bulbar muscular atrophy

**Subject Categories** Chromatin, Epigenetics, Genomics & Functional Genomics; Neuroscience

## Introduction

Spinal and bulbar muscular atrophy (SBMA) is an X-linked, intractable, adult-onset neurodegenerative disease caused by the expansion of a CAG repeat within the first exon of the *androgen receptor* (*AR*) gene (Kennedy *et al*, 1968; Sobue *et al*, 1989; Finsterer & Soraru, 2016), and the toxic function gained from the mutant AR in both spinal motor neurons and skeletal muscle underlies the pivotal SBMA pathomechanism (Katsuno *et al*, 2012a,b; Giorgetti & Lieberman, 2016; Pennuto & Basso, 2016). Accumulation of the pathogenic AR protein induces cellular damage and death by inducing biological deficits, including transcriptional dysregulation, axonal transport disruption, and mitochondrial dysfunction (Katsuno *et al*, 2012a,b). While transcriptional dysregulation is known to disrupt the expression of multiple genes, leading to crucial neuronal damage (Katsuno *et al*, 2010a,b; Minamiyama *et al*, 2012; Okazawa, 2017), the mechanisms underlying this pathological process have yet to be elucidated.

Polyglutamine-expanded proteins interfere with the activity of nuclear proteins possessing histone acetyltransferase (HAT) via abnormal protein interaction, which leads to transcriptional dysregulation in neurons (Suqars & Rubinsztein, 2003). This process has been confirmed by several lines of evidence, including the sequestration of transcriptional factors with HAT into the nuclear accumulation of polyglutamine-expanded proteins as well as the therapeutic effects of histone deacetylase inhibitors in various animal models of polyglutamine diseases (Butler & Bates, 2006). In addition to acetylation, altered monoubiquitination and methylation patterns of histone proteins have also been reported in a mouse model of Huntington's disease (McFarland *et al*, 2013; Vashishtha *et al*, 2013). However, because transcriptionally dysregulated genes involved in polyglutamine-mediated neurodegeneration are diverse, attributing this highly impactful molecular event to only altered histone

1 Department of Neurology, Nagoya University Graduate School of Medicine, Nagoya, Japan
2 Department of Neurology, University of Occupational and Environmental Health School of Medicine, Kitakyushu, Japan
3 Division of Cancer Biology, Nagoya University Graduate School of Medicine, Nagoya, Japan
4 Research Division of Dementia and Neurodegenerative Disease, Nagoya University Graduate School of Medicine, Nagoya, Japan
*Corresponding author. Tel: +81 52-744-2794; Fax: +81 52-731-3131; E-mail: sobueg@med.nagoya-u.ac.jp
**Corresponding author. Tel: +81 52-744-2391; Fax: +81 52-744-2394; E-mail: ka2no@med.nagoya-u.ac.jp
†These authors contributed equally to this work

modification is difficult, and other epigenetic dysregulations are likely implicated in the pathogenesis of polyglutamine diseases (Bassi et al, 2017).

DNA methylation, a gene silencing mechanism that does not affect the DNA sequence, is implicated in various biological processes, such as differentiation and imprinting (Smith & Meissner, 2013), and CpG islands in gene promoter regions are prone to methylation for a wide variety of reasons (Hirabayashi & Gotoh, 2010). While DNA methylation is a basic mechanism underlying healthy biological processes that respond to environmental stimuli, the abnormal DNA hyper-methylation of specific genes with CpG-rich promoters plays a fundamental role in pathological conditions, particularly malignancies (Jones & Baylin, 2007). For example, the tumor suppressor genes RB, MLH1, and BRCA1 are silenced by DNA hyper-methylation in the CpG islands of their promoter regions despite global DNA methylation suppression (Ferres-Marco et al, 2006; Stefansson et al, 2011; Muzny et al, 2012). Given that adult-onset neurodegenerative diseases have a considerably long latent period, they appear to be affected by environmental factors, similar to cancer (Torre et al, 2015). Because DNA methylation is also a key mechanism underlying epigenetic regulation of gene transcription in neurons (Feng et al, 2007; De Jager et al, 2014), we hypothesized that aberrant DNA methylation in specific genes with CpG-rich promoters is an underlying cause of transcriptional dysfunction in SBMA motor neurons.

Two critical postulations were evaluated to test this hypothesis. First, we studied whether DNA hyper-methylation due to aberrant expression of DNA methyltransferase (DNMT) is present in the spinal motor neurons of both mice and humans with SBMA. Second, we examined the effects of a drug that modulates DNA methylation in cellular and animal models of SBMA. We found that DNA methyltransferase 1 (Dnmt1) was highly expressed in the spinal motor neurons of SBMA mice and in patients with SBMA. RG108, a DNMT inhibitor, ameliorated the neurodegeneration induced by polyglutamine-expanded AR both in cultured neuronal cells and in mice by restoring the transcription of several genes, including Hes5. Together, these findings suggest that altered DNA methylation is an epigenetic pathomechanism underlying SBMA and a potential therapeutic target.

## Results

### Dnmt1 is highly expressed in the spinal cord motor neurons of SBMA mice

To clarify whether DNA methylation, an epigenetic modification that silences gene expression, influences transcriptional dysfunction in SBMA motor neurons, we first investigated DNA methylation alteration in an SBMA mouse model bearing human AR with 97 CAG repeat sequences (AR-97Q; Katsuno et al, 2002, 2003; Adachi et al, 2005). This model organism has prominent neurodegenerative features, including muscle atrophy and weakness with a late onset at approximately 10 weeks of age as well as nuclear pathogenic AR accumulation in affected neurons, all of which are similar to the features seen in affected humans (Atsuta et al, 2006). We examined the expression levels of 3 DNMTs, namely Dnmt1, Dnmt3a, and Dnmt3b, that are key enzymes regulating DNA methylation in

specific genes, including tumor suppressor genes. Western blot analysis revealed an intensified protein level of Dnmt1 and unaltered level of Dnmt3a and Dnmt3b in the spinal cords of AR-97Q mice compared with wild-type and AR-24Q mice bearing normal-sized CAG repeats (Fig 1A and B). Using RT–qPCR, we confirmed that Dnmt1 transcription was up-regulated in the spinal cords of AR-97Q mice (Fig 1C). We also performed immunohistochemistry to clarify the cellular localization of DNMTs in the spinal anterior horn motor neurons of AR-97Q mice. While Dnmt3a and Dnmt3b were mainly stained in the cytoplasm of the motor neurons in wild-type, AR-24Q, and AR-97Q mouse spinal cords, Dnmt1 immunoreactivity was enriched in the nucleus of spinal motor neurons in SBMA model mice (Fig 1D and E). Immunofluorescence staining verified the co-localization of Dnmt1 with 1C2, an abnormal polyglutamine marker, in the spinal motor neurons of SBMA mice; this phenomenon was observed in 85.7% of neurons (Fig 1F). In addition, we investigated the protein levels of Dnmt1 in central nervous system regions other than the spinal cord as well as in several non-neuronal organs, including skeletal muscle, liver, and testes, in AR-97Q mice. Dnmt1 protein level in skeletal muscle, wherein mutant AR exerts toxicity, was not different between wild-type and AR-97Q mice (Fig 1G and H, and Appendix Fig S1). Moreover, in the liver, testis, cerebellum, and cerebral cortex, Dnmt1 level was similar between wild-type and AR-97Q mice (Appendix Figs S2 and S3). Taken together, these findings suggest that Dnmt1 up-regulation is associated with the nuclear accumulation of polyglutamine-expanded AR in affected spinal motor neurons. We also investigated the localization of 5-methylcytosine (5mC), a marker of whole DNA methylation, using immunohistochemistry. We found the localization of 5mC in the spinal motor neurons and in the skeletal muscle of AR-97Q mice was nucleus-dominant, which was similar to that of wild-type mice (Appendix Figs S4 and S5). To reveal whether the Dnmt1 of spinal motor neurons of SBMA patients has similar alterations to that of SBMA model mice, we compared the localization and immunoreactivity of Dnmt1 in post-mortem spinal cord samples from diseased patients and controls. Dnmt1 was enriched in the nucleus of spinal motor neurons in SBMA patients, similar to the immunoreactivity observed in SBMA model mice (Appendix Fig S6).

### Genetic and pharmacological Dnmt1 suppression ameliorates SBMA model cells

To clarify whether Dnmt1 up-regulation has a causative role in SBMA neuronal damage, we analyzed the function of Dnmt1 in cultured neuronal cells bearing the pathogenic polyglutamine-expanded AR. To this end, we used differentiated NSC34 mouse motor neuron-like cells stably expressing human AR containing 97 glutamines (NSC97Q). As reported previously, in these cells, abnormal AR protein accumulates in the nucleus in a testosterone-dependent manner, which is a pathological feature of SBMA (Minamiyama et al, 2004; Miyazaki et al, 2012). In the NSC34 cells bearing AR-97Q that were treated with dihydrotestosterone (DHT), Dnmt1 expression was up-regulated compared to cells carrying human AR with a normal polyglutamine tract (AR-24Q) with DHT, while that of Dnmt3a and Dnmt3b was not changed (Fig 2A); the same phenomena were observed in spinal cord lysates of SBMA mice, as determined by Western blot and RT–qPCR (see Fig 1A–C). Furthermore, significant enhancement of the Dnmt1 expression

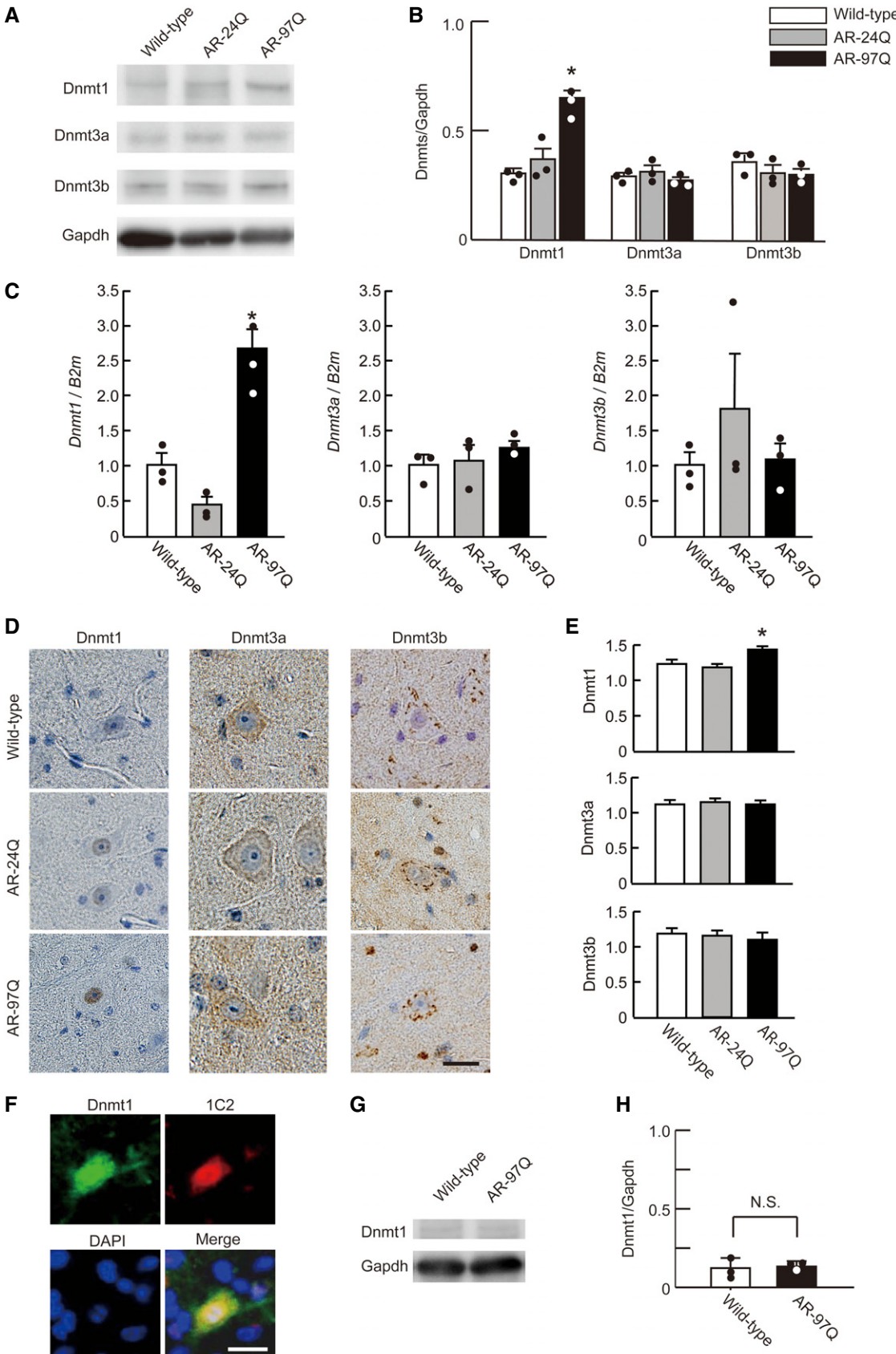

Figure 1.

**Figure 1. DNMT level in the spinal motor neurons of SBMA mice.**

A  Immunoblotting for Dnmt1, Dnmt3a, and Dnmt3b in the spinal cords of AR-97Q, AR-24Q, and wild-type mice.
B  Quantitative densitometry analysis indicated that Dnmt1 spinal cord level was up-regulated in AR-97Q mice ($n = 3$). Dnmt3a and Dnmt3b spinal cord level was not significantly different among wild-type, AR-24Q, and AR-97Q mice ($n = 3$).
C  Quantification of the *Dnmts* mRNA levels of the spinal cord in wild-type, AR-24Q, and AR-97Q mice using RT–qPCR ($n = 3$).
D, E  Representative immunohistochemical images depicting the expression of Dnmt1, Dnmt3a, and Dnmt3b in the spinal cords of male AR-97Q, AR-24Q, and wild-type mice and quantification of the immunoreactivity for Dnmts ($n = 25$ neurons from three mice of each group).
F  Double-immunofluorescence staining for Dnmt1 and 1C2 in the spinal motor neurons of an AR-97Q mouse.
G  Immunoblotting for Dnmt1 of the skeletal muscle of wild-type and AR-97Q mice.
H  Quantification of the signal intensity of Dnmt1 bands ($n = 3$).

Data information: One-way ANOVA with Tukey's test (B, C, E). Unpaired *t*-test (H). Error bars, s.e.m. Scale bar, 20 μm (D, F). \*$P < 0.05$. N.S.: not significant. 1C2, polyglutamine. DAPI, 4′,6-diamidino-2-phenylindole. The exact *P*-value is in Appendix Table S3.
Source data are available online for this figure.

level was observed in DHT-treated NSC97Q cells compared with untreated NSC97Q cells (Fig 2B). These findings were not found in DHT-untreated NSC97Q cells (Fig 2C and D). To clarify the role of DNMTs in SBMA pathology, we down-regulated DNMTs in the DHT-treated NSC97Q cells using siRNA. While *Dnmt1* knockdown improved the NSC97Q cell viability, *Dnmt3a* and *Dnmt3b* depletion had no such effect (Fig 2E and Appendix Fig S7). Additionally, knockdown of *Dnmts* had no effect on cell viability in DHT-untreated NSC97Q cells or in DHT-treated NSC24Q cells (Fig 2F, and Appendix Figs S8 and S9). We next administered RG108, a DNMT inhibitor, to the NSC97Q cells that were treated with DHT. RG108 ameliorated the SBMA model cell viability in a dose-dependent manner (Fig 2G). Although DNMTs' levels were not affected by low doses of RG108, higher concentrations (1 and 10 μM) of RG108 reduced the expression of Dnmt1 but not that of Dnmt3a or Dnmt3b, similar to that reported previously in cancer cells (Savickiene *et al*, 2012; Gracia *et al*, 2014), suggesting that RG108 is relatively specific to Dnmt1 in the NSC97Q cells (Fig 2H). RG108 did not change the cell viability of DHT-untreated NSC97Q cells or DHT-treated NSC24Q cells as evaluated with WST-8 assay (Fig 2I and J).

**RG108 mitigates SBMA *in vivo***

To examine whether RG108 has a similar therapeutic effect *in vivo*, we administered the agent to AR-97Q mice. Because altered DNA methylation was observed only in spinal motor neurons and not in visceral organs, including muscle, we continuously infused RG108 intracerebroventricularly for 2 weeks using an osmotic pump into

6-week-old AR-97Q mice, an age at which the disease is presymptomatic. Behavior analysis, including grip power, body weight, and a rotarod task, showed that RG108 ameliorated the motor function of SBMA mice (Fig 3A–C). Moreover, RG108 significantly improved the survival rate (Fig 3D) and mitigated the muscle atrophy of AR-97Q mice (Fig 3E). Dnmt1 level was significantly suppressed in RG108-treated SBMA mice compared with DMSO-treated SBMA mice, as determined by Western blot (Fig 3F and G). RG108 had no significant effects on the motor functions or survival rates of wild-type mice (Appendix Fig S10). To further clarify whether RG108 changes the localization of Dnmt1 expression in the spinal motor neurons of AR-97Q mice, we performed immunohistochemistry on DMSO- and RG108-treated mice. The localization of Dnmt1 was not altered, but immunoreactivity was attenuated, in the motor neurons of RG108-treated mice compared with those of the control group (Fig 3H and I).

**RG108 ameliorates motor neuron degeneration without suppressing AR accumulation**

To assess whether RG108 therapy suppresses pathogenic AR accumulation in SBMA mice, we performed immunohistochemistry using an anti-polyglutamine antibody. RG108 had virtually no effects on AR accumulation in the spinal motor neurons of AR-97Q mice (Fig 4A and B), and pathogenic AR protein levels were also not decreased, as determined by immunoblotting (Fig 4C and D). RT–qPCR showed that the expression of human *AR* was not altered (Fig 4E). To clarify whether the therapeutic effects of RG108 on survival and motor function are dependent on the suppression of

**Figure 2. Dnmt1 suppression improves the cell viability of neuronal SBMA cells.**

A  Immunoblotting of NSC34 cells expressing AR-24Q or AR-97Q (NSC24Q and NSC97Q, respectively) that were treated with DHT for Dnmt1, Dnmt3a, and Dnmt3b, and quantification of the signal intensities of bands corresponding to Dnmt1, Dnmt3a, and Dnmt3b ($n = 3$).
B  Relative mRNA levels of *Dnmt1* in NSC24Q and NSC97Q cells measured with RT–qPCR ($n = 3$).
C  Protein levels of Dnmt1, Dnmt3a, and Dnmt3b in DHT-treated and DHT-untreated NSC97Q cells by Western blotting analysis.
D  RT–qPCR analysis of *Dnmt1* in DHT-treated and DHT-untreated NSC97Q cells ($n = 3$).
E, F  Quantitative assessment of cell viability changes due to the siRNA-mediated knockdown of *Dnmt1*, *Dnmt3a*, or *Dnmt3b* in NSC97Q cells treated with DHT (E) or without DHT (F) using the WST-8 assay ($n = 3$).
G  Quantitative assessment of the cell viability of NSC97Q cells treated with DHT and RG108 using the WST-8 assay ($n = 3$ per group).
H  Immunoblotting of NSC97Q cells treated with DHT and RG108 for Dnmt1, Dnmt3a, and Dnmt3b.
I, J  Cell viability analysis with siRNA-mediated knockdown of *Dnmts* in DHT-untreated NSC97Q (I) and DHT-treated NSC24Q (J).

Data information: Unpaired *t*-test. Error bars, s.e.m. \*$P < 0.05$, N.S.: not significant. The exact *P*-value is in Appendix Table S3.
Source data are available online for this figure.

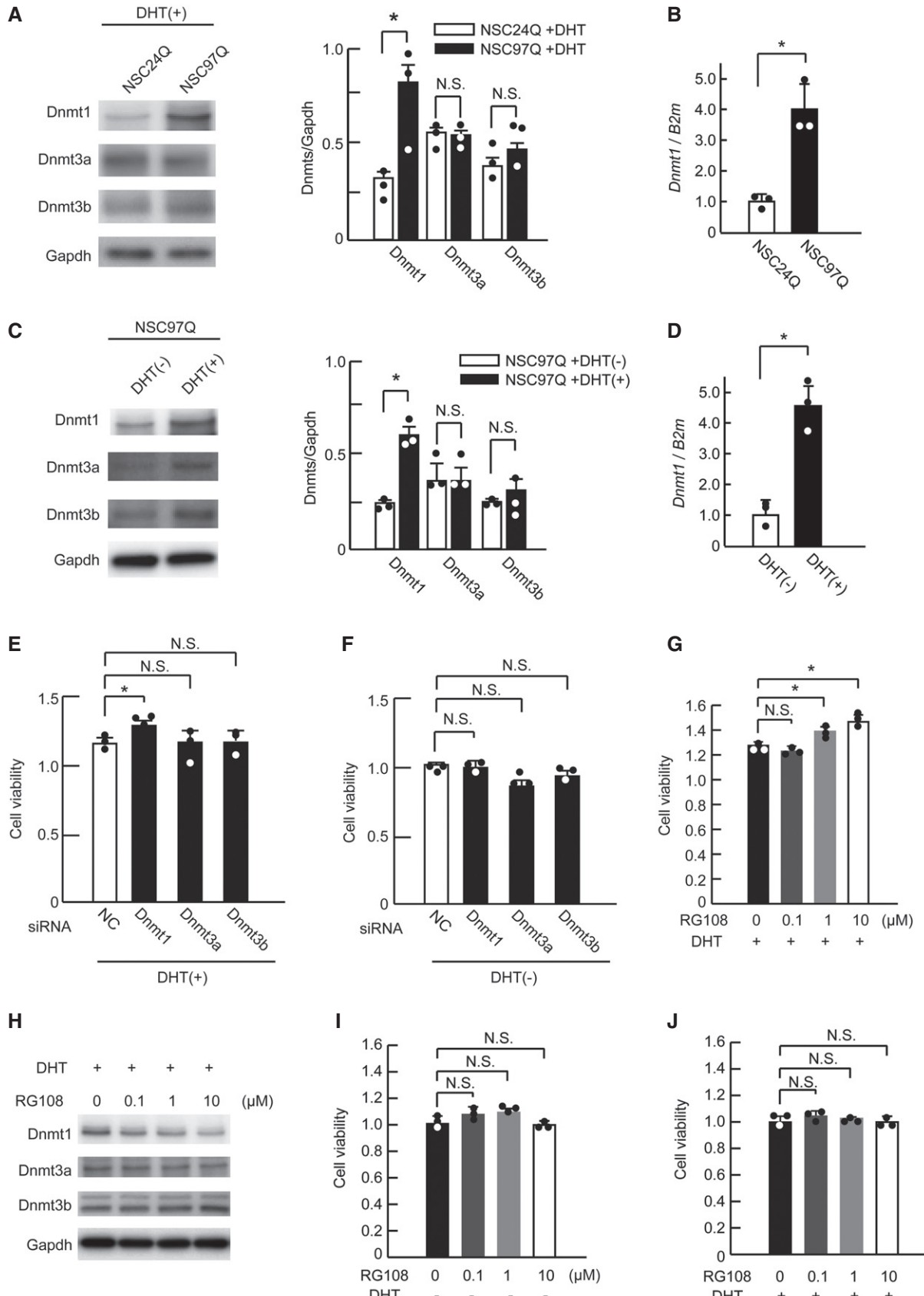

Figure 2.

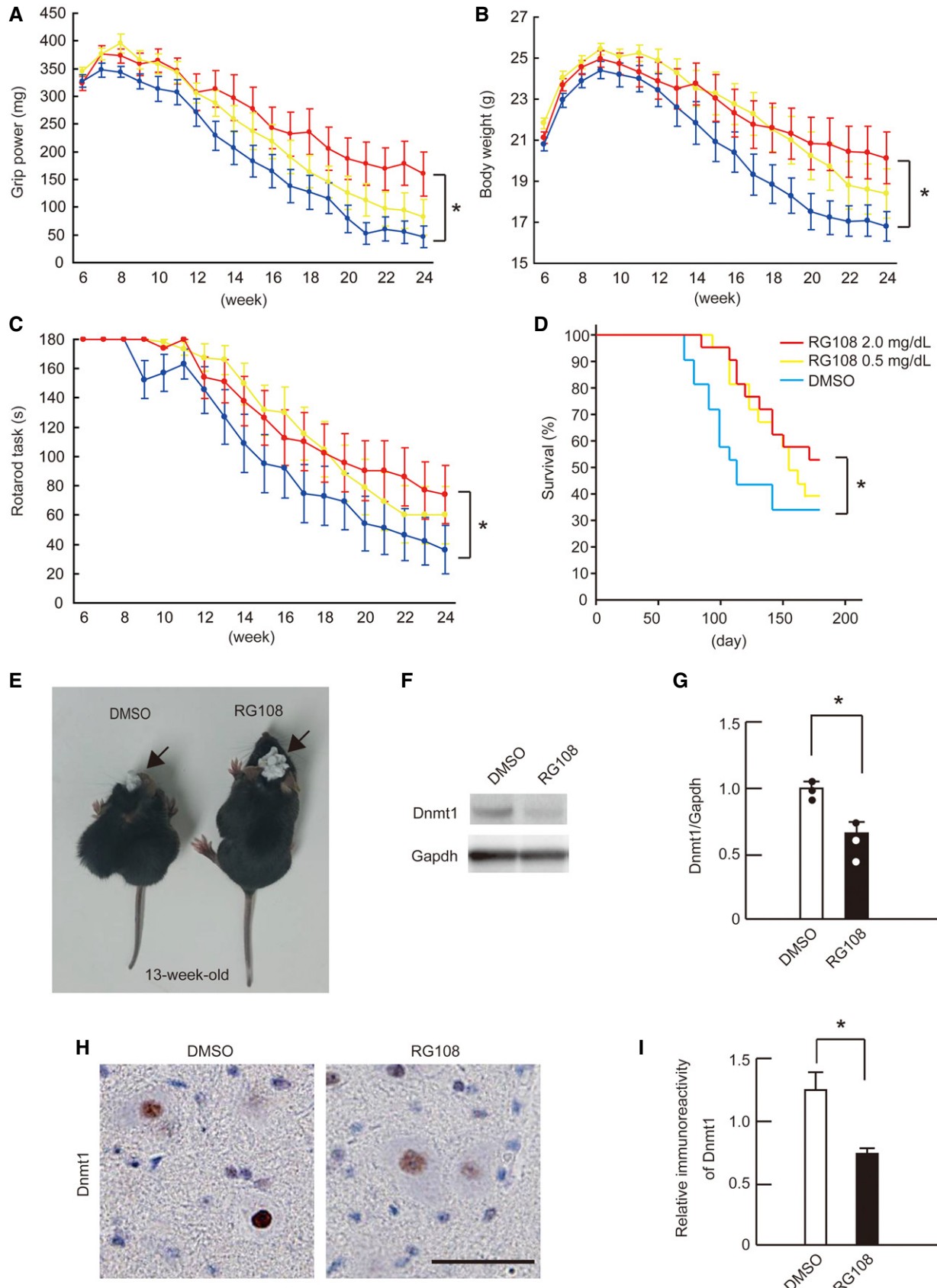

Figure 3.

**Figure 3. RG108 improves the motor function and survival of SBMA mice.**

A–D   Grip power (A), body weight (B), rotarod performance (C) (two-way ANOVA with Tukey's test), and survival rate (D) (Kaplan–Meier analysis and log-rank test) of AR-97Q mice treated with or without RG108 (DMSO, *n* = 20; 0.5 mg/dl, *n* = 20; 2.0 mg/dl, *n* = 20). All parameters were significantly improved in AR-97Q mice treated with 2.0 mg/dl RG108 compared with those treated with DMSO: **P* < 0.05 (grip); **P* < 0.05 (body weight); **P* < 0.05 (rotarod); and **P* < 0.05 (survival).

E     Representative photograph of a 13-week-old AR-97Q mouse treated with DMSO (left) and an age-matched AR-97Q mouse treated with 2.0 mg/dl RG108 (right). AR-97Q mouse medicated with RG108 maintained muscle volume and body size compared with DMSO-treated AR-97Q mouse of the same age. The white compact masses (arrow) of mice head are the cement covering the surgical site.

F, G   Immunoblots (F) and relative intensities (G) of the Dnmt1-immunoreactive bands of spinal cords from AR-97Q mice treated with or without RG108 (*n* = 3 per group).

H, I   Dnmt1 immunohistochemistry in spinal cord sections from AR-97Q mice treated with or without RG108 (*n* = 3 per group).

Data information: Two-way ANOVA with Tukey's test (A–C). Log-rank test (D). Unpaired *t*-test (G, I). Error bars, s.e.m. Scale bar, 20 μm (H). **P* < 0.05. The exact *P*-value is in Appendix Table S3.

Source data are available online for this figure.

motor neuron degeneration in the spinal anterior horn, we examined the protein levels of ChAT, a histological marker of motor neurons. RG108 suppressed spinal motor neuron atrophy in AR-97Q mice (Fig 4F and G), and ChAT protein levels were elevated in the spinal cords of RG108-treated SBMA mice compared with their DMSO-treated counterparts (Fig 4H and I).

## Hes5 is silenced via promoter methylation in SBMA cells

To examine whether Dnmt1 up-regulation in SBMA results in increased DNA methylation in the CpG island region of the promoter, which controls the expression of downstream genes, we performed DNA methylation array analysis using human neuroblastoma SH-SY5Y cells stably expressing human AR containing 24 or 97 glutamine residues (SH24Q, SH97Q), treated with DHT. First, we examined the protein expression levels of DNMTs in DHT-treated SH24Q and SH97Q cells using Western blots; the results confirmed that the protein level of Dnmt1, but not Dnmt3a or Dnmt3b, was higher in the SH97Q cells than in the SH24Q cells (Fig 5A and B). We confirmed that the up-regulation of Dnmt1 is testosterone-dependent in SH97Q cells (Fig 5C and D). DNA methylation array analysis using these cells revealed that DNA methylation of CpG islands is intensified in several genes. However, total DNA methylation level was not altered between SH24Q and SH97Q (Appendix Fig S11). Of the top 50 genes with hyper-methylated CpG islands in their promoter region (Appendix Tables S1 and S2), seven genes were identified that reportedly function in neurons: *CDC25B*, *GFRA3*, *NPY*, *HES5*, *SCTR*, *LEF1-AS1*, and *CABS1* (Fig 5E). To determine whether these candidate genes were expressed at lower levels in DHT-treated SH97Q cells than in SH24Q cells treated with the same hormone, we analyzed their mRNA expression with RT–qPCR, showing that *HES5* was most silenced among the candidates (Fig 5F). Furthermore, DHT treatment reduced the *Hes5* mRNA level in SH97Q cells (Fig 5G).

To determine whether *HES5* expression is controlled by DNA methylation of the CpG island of its promoter region, we performed methylation-specific PCR analysis (MSP). Methylation of the *HES5* promoter CpG island in DHT-treated SH97Q cells was higher than that of DHT-treated SH24Q cells (Fig 5H). Hyper-methylation of the *HES5* promoter CpG island was also testosterone-dependent in SH97Q cells (Appendix Fig S12). To confirm that the same phenomenon was reproduced in a murine SBMA neuronal cell model, we performed MSP using NSC97Q cells that were treated with DHT, and the *Hes5* promoter region was found to be hyper-methylated in

NSC34 cells bearing pathogenic AR. To examine whether the increased DNA methylation of the *Hes5* CpG island silenced its gene expression, we measured *Hes5* mRNA levels in NSC97Q cells. As seen in the SH97Q cells, *Hes5* mRNA expression was suppressed in DHT-treated NSC97Q cells (Appendix Fig S13). To assess the localization of Hes5 protein in SBMA, we performed immunohistochemistry on the spinal cords of AR-97Q mice. Hes5 immunoreactivity was attenuated in the motor neurons of SBMA mice (Fig 5I), and *Hes5* mRNA expression was significantly reduced in the spinal cords of AR-97Q mice compared with that of wild-type mice and AR-24Q mice (Fig 5J).

## RG108 recovers Hes5 expression

To clarify whether RG108 alters both the promoter DNA methylation and mRNA expression of *Hes5*, we performed MSP and RT–qPCR analysis using SH97Q and NSC97Q cells that were treated with DHT. RG108 reduced DNA methylation in the *Hes5* promoter region and thereby elevated the *Hes5* mRNA levels in the SH97Q cellular model of SBMA (Fig 6A and B). Similar phenomena were observed in the NSC97Q SBMA cell model, as DNA methylation of the *Hes5* promoter region was suppressed and the *Hes5* mRNA levels were restored by RG108 treatment (Fig 6C and D). The cell viability of DHT-untreated SH97Q and NSC97Q cells was not affected by RG108 (Appendix Fig S14). Furthermore, RG108 treatment ameliorated the *Hes5* mRNA levels in the spinal cord in this disease model (Fig 6E); immunoreactivity of Hes5 was localized in the nucleus and cytoplasm of motor neurons in the spinal cords of AR-97Q mice, and it was diminished by RG108 treatment (Fig 6F and G).

## Hes5 prevents motor neuron degeneration in SBMA cells

Hes5 is known as a key molecule in neuronal development (Hatakeyama *et al*, 2004, 2006); while Hes5 and Hes1 are reportedly crucial for regulating embryonic stem cell differentiation (Kageyama *et al*, 2008; Imayoshi *et al*, 2010), the role of Hes5 in mature neurons is unclear. To clarify whether Hes5 plays a protective role in the polyglutamine-induced neurodegeneration of SBMA, we examined the effect of *Hes5* siRNA knockdown. *Hes5* depletion deteriorated the viability of DHT-treated NSC97Q cells, as measured by the WST-8 assay (Fig 7A). A marked decline in the constitutive protein levels of Hes5 was confirmed with Western blotting (Fig 7B). To confirm that RG108 does not exert benefits in the

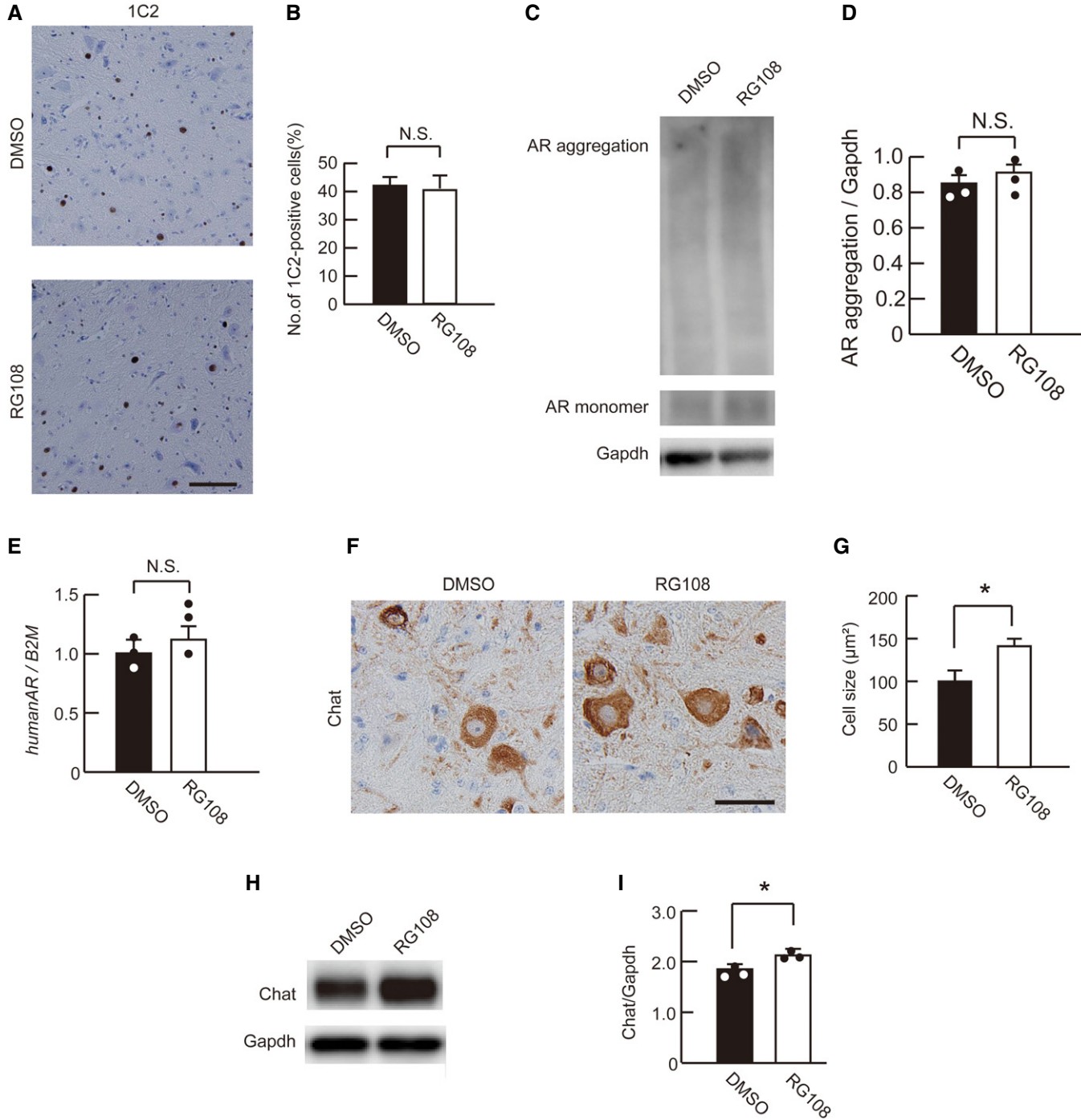

**Figure 4. RG108 attenuates motor neuron degeneration in SBMA mice without degrading the disease-causing abnormal AR protein.**

A   Immunohistochemistry of the spinal cords of AR-97Q mice treated with DMSO or RG108 for polyglutamine using a 1C2 antibody.

B   Quantification of 1C2-positive motor neurons in the spinal cords (*n* = 5 per group).

C   Immunoblotting of the spinal cords of AR-97Q mice treated with or without RG108 for AR.

D   Densitometric analyses to quantify AR accumulation in the spinal motor neurons of AR-97Q mice treated with DMSO or RG108 (*n* = 3).

E   Relative mRNA expression levels of human *AR* in AR-97Q mouse spinal cords with or without RG108 treatment (*n* = 3).

F, G   Immunohistochemistry for ChAT (F) and quantitative analysis of the neuron sizes (G) in the spinal cord anterior horns of AR-97Q mice (*n* = 5 per group).

H, I   Western blot analysis (H) and relative signal intensities (I) of the ChAT-immunoreactive bands of spinal cords from AR-97Q mice treated with or without RG108 (*n* = 3).

Data information: Unpaired *t*-test. Error bars, s.e.m. Scale bar, 20 μm (A, F). *P < 0.05. N.S.: not significant. The exact *P*-value is in Appendix Table S3.
Source data are available online for this figure.

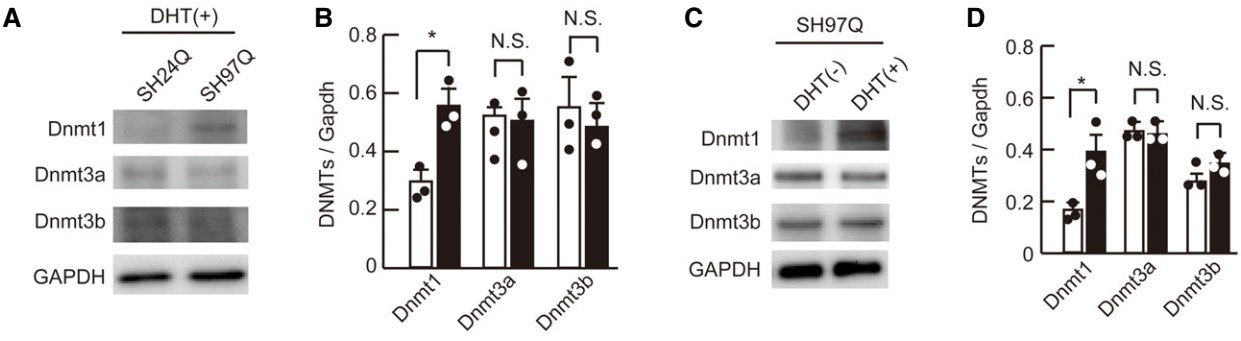

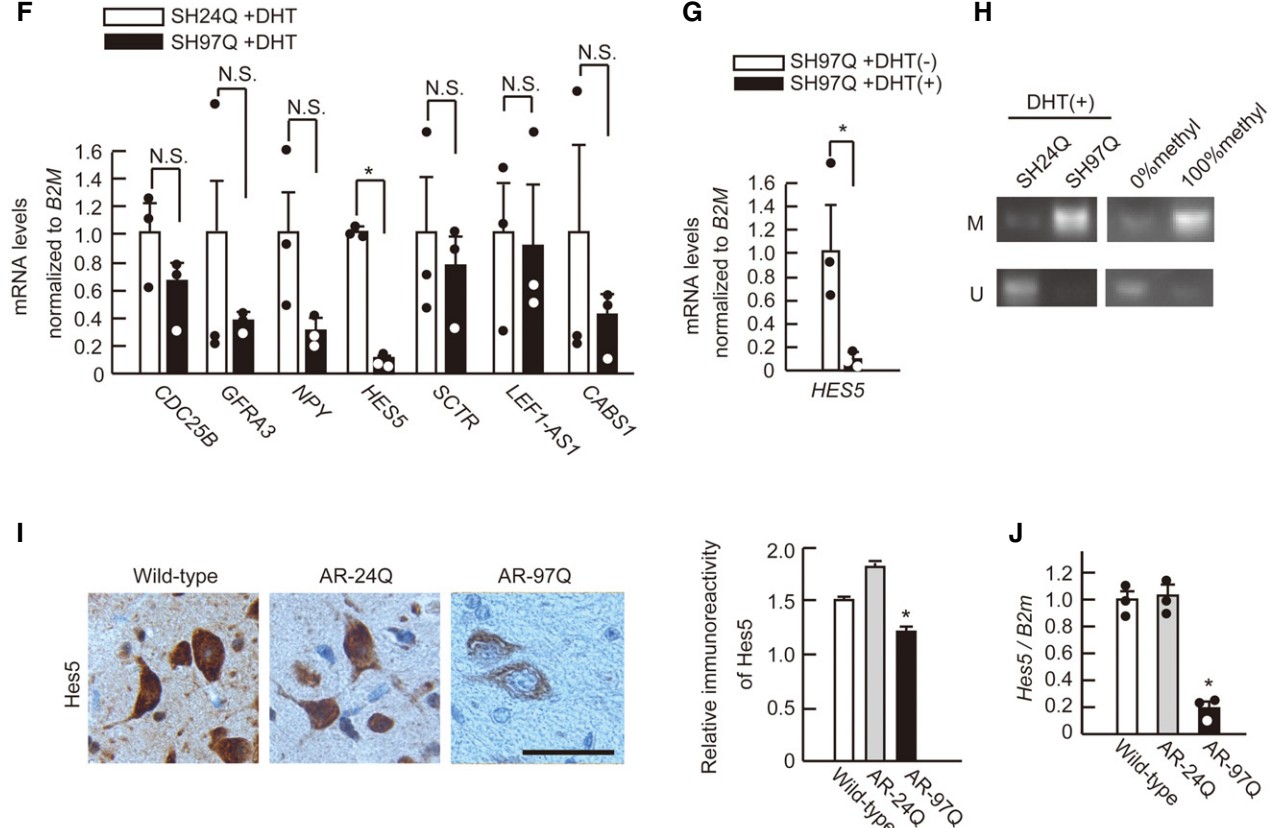

**Figure 5.**

◀

**Figure 5. DNA methylation array analysis identified Hes5 as a key molecule for SBMA pathogenesis.**

A  Immunoblotting of SH-SY5Y cells stably expressing AR-24Q (SH24Q) or AR-97Q (SH97Q) for Dnmt1, Dnmt3a, and Dnmt3b.
B  Densitometry quantification showing that Dnmt1 level was up-regulated in SH97Q cells compared with SH24Q cells (n = 3).
C, D  Immunoblots for Dnmts of SH97Q with or without DHT treatment (C) and quantification data (D) (n = 3).
E  Seven genes with DNA hyper-methylation in their CpG islands were identified by DNA methylation array analysis.
F  The relative mRNA levels of *CDC25B*, *GFRA3*, *NPY*, *HES5*, *SCTR*, *LEF1-AS1*, and *CABS1*, normalized to *beta-2 microglobulin*, in DHT-treated SH24Q and SH97Q cells were measured using RT–qPCR (n = 3).
G  RT–qPCR analysis indicated the decrease of *Hes5* mRNA level in DHT-treated SH97Q cells compared with DHT-untreated SH97Q cells (n = 3).
H  Methylation-specific PCR of SH24Q and SH97Q cells that were treated with DHT.
I  Hes5 immunoreactivity of spinal motor neurons in wild-type, AR-24Q, and AR-97Q mice (n = 3).
J  *Hes5* mRNA levels in the spinal cords of wild-type, AR-24Q, and AR-97Q mice (n = 3).

Data information: Unpaired *t*-test (B, D, F, G). One-way ANOVA with Tukey's test (I, J). Error bars, s.e.m. Scale bar, 50 μm (I). *$P < 0.05$. N.S.: not significant. The exact *P*-value is in Appendix Table S3.
Source data are available online for this figure.

absence of *Hes5*, we assessed the cell viability of *Hes5*-knocked-down NSC97Q cells treated with DHT and RG108, and the therapeutic effects of RG108 were counteracted by siRNA *Hes5* knockdown (Fig 7C). On the other hand, Hes5 over-expression improved the cell viability of the cellular SBMA model (Fig 7D and E). We also confirmed the protective effects of exogenous Hes5 in NSC97Q cells using the lactate dehydrogenase (LDH) assay (Fig 7F). To examine whether Hes5 over-expression suppresses pathogenic AR accumulation, we performed Western blot analysis on the NSC97Q cells that were treated with DHT using an anti-AR antibody. The pathogenic AR protein levels, both the high-molecular weight protein complex and the monomer, were not changed by Hes5 over-expression *per se* (Fig 7G and H). Taken together, these results suggest that Hes5 can protect the neuronal SBMA cellular model without suppressing pathogenic AR accumulation. The siRNA-mediated knockdown of *Hes5* induced cell damage in DHT-untreated NSC97Q cells, although RG108 had no effect in those cells (Appendix Fig S15A–C). Over-expression of Hes5 increased the cell viability of DHT-untreated NSC97Q cells (Appendix Fig S15D–F). These findings indicated that Hes5 has a positive effect on the viability of normal cells as well as SBMA model cells.

We further explored how Hes5 mitigates the neurodegenerative process. Since cellular signaling pathways including TGF-β, NFκB, and heat shock proteins have been shown to play crucial roles in the pathogenesis of SBMA (Katsuno *et al*, 2010b; Kondo *et al*, 2013; Iida *et al*, 2015), we performed siRNA-mediated *Hes5* knockdown in NSC34 cells and examined the levels of key proteins for such signaling. *Hes5* depletion suppressed Smad2 phosphorylation but had no effect on heat shock factor-1 (Hsf1) or IκBα (Fig 8A–C). Phosphorylation of Smad2 was suppressed in DHT-treated NSC97Q cells (Fig 8D), and this phenomenon was reversed by Hes5 over-expression (Fig 8E). Moreover, we confirmed that AR-97Q decreased the level of Hes5 and Smad2 phosphorylation in primary cortical neurons (Fig 8F and G) as well as in primary motor neurons (Fig 8H). Lentiviral vector-mediated over-expression of Hes5 restored phosphorylation of Smad2 in primary cortical neurons expressing AR-97Q (Fig 8I).

## Discussion

This study demonstrated that Dnmt1 is up-regulated and plays a causative role in the neurodegeneration of SBMA. The relationship between DNA methylation and neuronal function has been exemplified in Rett syndrome, an autism spectrum disorder with extrapyramidal signs. In this disease, because MeCP2, a key protein suppressing gene expression via DNA methylation, is not capable of binding methylated DNA, DNA methylation goes unrecognized (Bienvenu & Chelly, 2006; Liyanage & Rastegar, 2014), clearly indicating that DNA methylation dysregulation leads to neuronal dysfunction. However, only fragmentary evidence exists that implicates DNA methylation in neurodegeneration (Sanchez-Mut *et al*, 2016). Here, we revealed that the hyper-methylation of certain genes by Dnmt1 is implicated in the neurodegenerative process of a polyglutamine-mediated disease. Although altered histone modifications have been extensively studied in various neurodegenerative diseases, including Huntington's disease and Alzheimer's disease (Sadri-Vakili & Cha, 2006; Fischer, 2014), our results provide additional insight that dysregulated DNA methylation is a potent epigenetic factor inducing neurodegeneration.

Although Dnmt1 is highly up-regulated in neurons bearing a polyglutamine-expanded AR, global DNA methylation levels were not changed in such cells. Instead, Dnmt1 induced the methylation of certain genes, including *Hes5*, in cellular and mouse models of SBMA. While this appears paradoxical, similar findings have been widely reported in cancer cells. DNA hyper-methylation in the promoters or enhancers of specific genes, such as cancer suppressor genes, without global DNA hyper-methylation is a characteristic feature of DNA methylation patterns in cancer. Rather, global DNA methylation is often attenuated due to hypo-methylation in DNA repeat sequences (Esteller, 2007). Thus, our results suggest that the altered DNA methylation in SBMA is similar to that in malignancies.

Our data also provide evidence that pharmacological suppression of DNA methylation in an SBMA model mouse can ameliorate the phenotype, including motor function and survival, suggesting that therapeutic intervention targeting epigenetics is effective for polyglutamine-mediated neurodegenerative disease model animals. Moreover, DNMT inhibitors, including RG108, 5-azacytidine (5-aza-dC), and zebularine, reportedly suppress the survival of various types of cancer cells (Brueckner *et al*, 2005; Yang *et al*, 2013; Gayet *et al*, 2015), and 5-aza-dC has indeed been widely used for treating myelodysplastic syndromes (MDS; Fenaux *et al*, 2009; Gore *et al*, 2013).

A limited number of reports illustrate the beneficial effects on neurodegeneration of suppressing DNA methylation. DNMT inhibition restored the expression of several key genes, including *Bdnf*, in

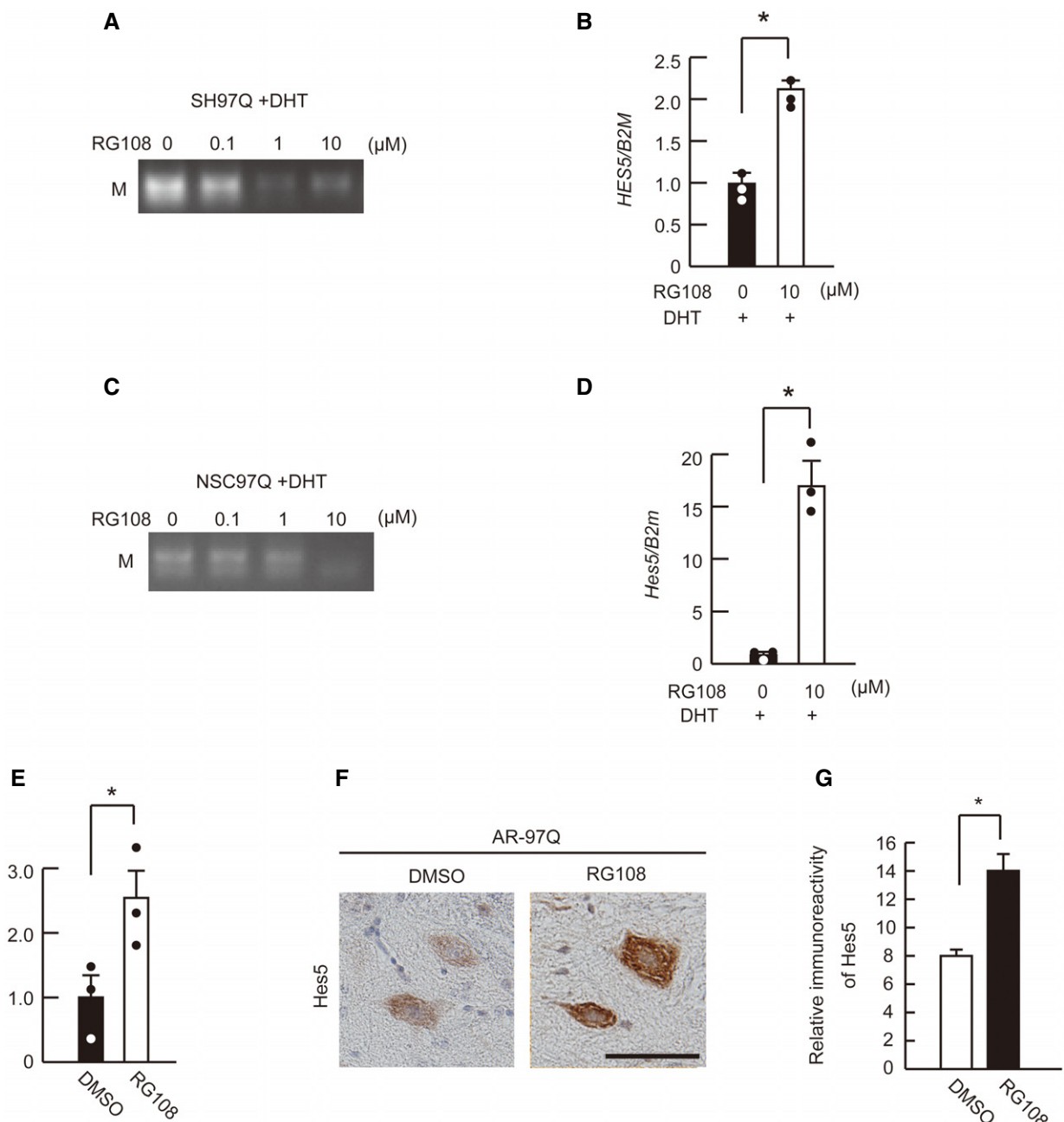

**Figure 6. RG108 restores Hes5 level by reducing DNA methylation.**

A, B  DNA methylation levels of the *HES5* promoter region (A) and the *HES5* mRNA levels (*n* = 3) (B) of SH97Q cells treated with DHT and RG108.

C, D  DNA methylation levels of the *Hes5* promoter region (C) and the mRNA levels of *Hes5* (*n* = 3) (D) in DHT-treated NSC97Q cells treated with or without RG108.

E  *Hes5* mRNA levels in the spinal cords of AR-97Q mice treated with or without RG108 (*n* = 3).

F, G  Hes5 immunoreactivity of spinal motor neurons in AR-97Q mice treated with or without RG108 (*n* = 3).

Data information: Unpaired *t*-test. Error bars, s.e.m. Scale bar, 20 μm (F). *P < 0.05. The exact *P*-value is in Appendix Table S3.

Source data are available online for this figure.

both a cellular and mouse model of Huntington's disease (Pan *et al*, 2016). Furthermore, locally injecting RG108 into the brain has been shown to attenuate the pathological accumulation of abnormal proteins in a mouse model of amyotrophic lateral sclerosis (ALS; Chestnut *et al*, 2011). However, the mechanism underlying such effects remains unclear. In the present study, we identified Hes5 as

the target of RG108. Hes1 and Hes5 are known to be notch target genes in neuronal cell differentiation (Kageyama & Ohtsuka, 1999), and although they regulate embryonic stem cell development via suppressing Notch signaling, their roles in mature neuronal cells are unclear. Interestingly, depletion of *Hes5* suppressed Smad2 activation in NSC34 cells, and over-expression of *Hes5* restored Smad2

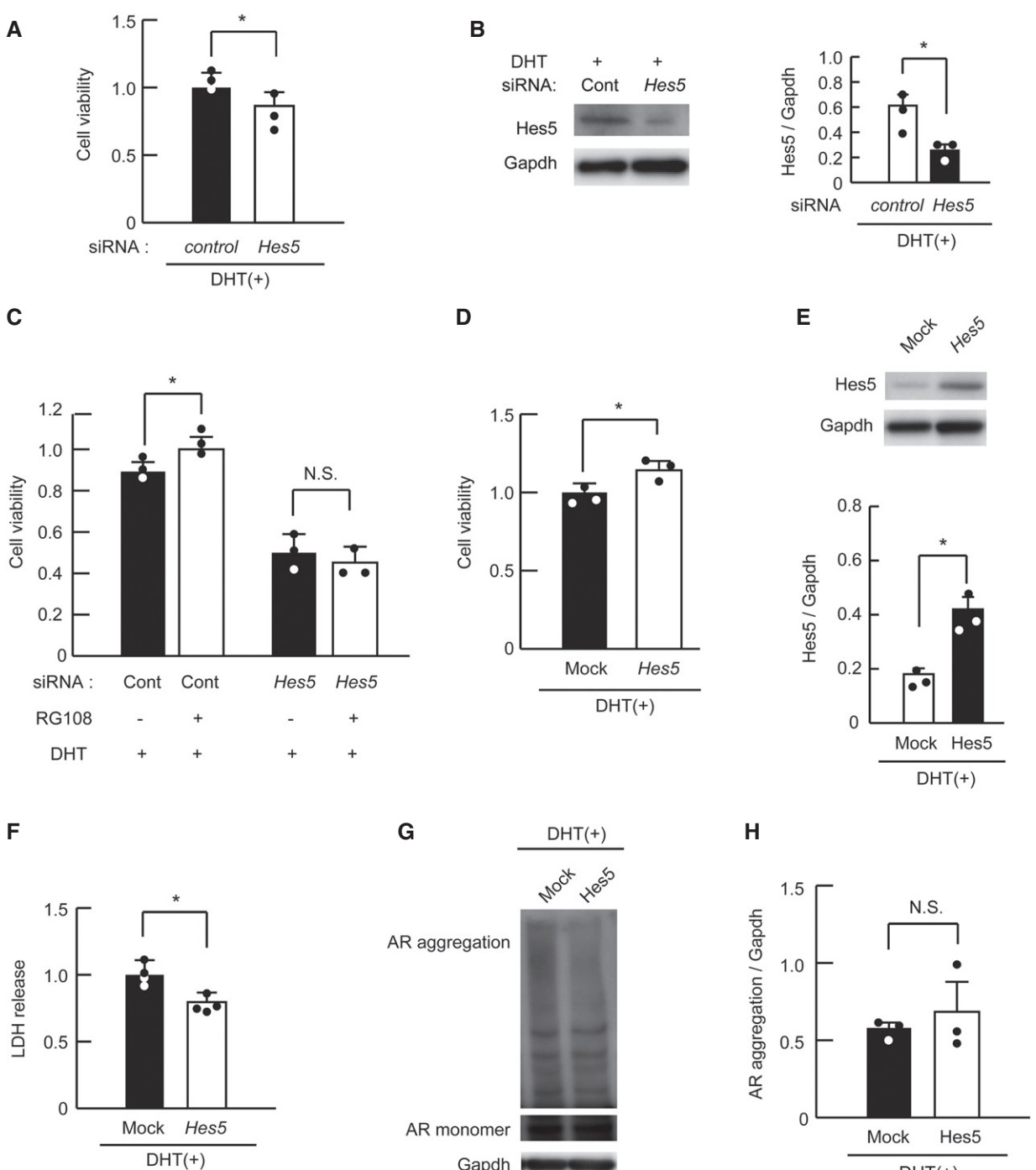

**Figure 7. Hes5 copes with pathogenic AR cytotoxicity.**

A    Viability of DHT-treated NSC97Q cells treated with *Hes5* and control siRNA measured by the WST-8 assay (*n* = 3).

B    Immunoblots of DHT-administered NSC97Q cells treated with *Hes5* siRNA (*n* = 3).

C    Quantitative cell viability analysis revealed that siRNA-mediated *Hes5* knockdown negated the therapeutic effect of RG108 in NSC97Q cells that were treated with DHT (*n* = 3), as determined by the WST-8 assay.

D, E   Cell viability measured with the WST-8 assay (D), and immunoblotting (E) of DHT-treated NSC97Q cells transfected with a mock plasmid or the *Hes5* vector (*n* = 3).

F    Quantification of LDH release from DHT-treated NSC97Q cells transfected with a mock plasmid or the *Hes5* vector (*n* = 4).

G    Immunoblots of DHT-administered NSC97Q cells transfected with a mock plasmid or the *Hes5* vector for AR.

H    AR aggregation as quantified by densitometry (*n* = 3).

Data information: Unpaired *t*-test. Error bars, s.e.m. *P < 0.05. N.S.: not significant. The exact *P*-value is in Appendix Table S3.
Source data are available online for this figure.

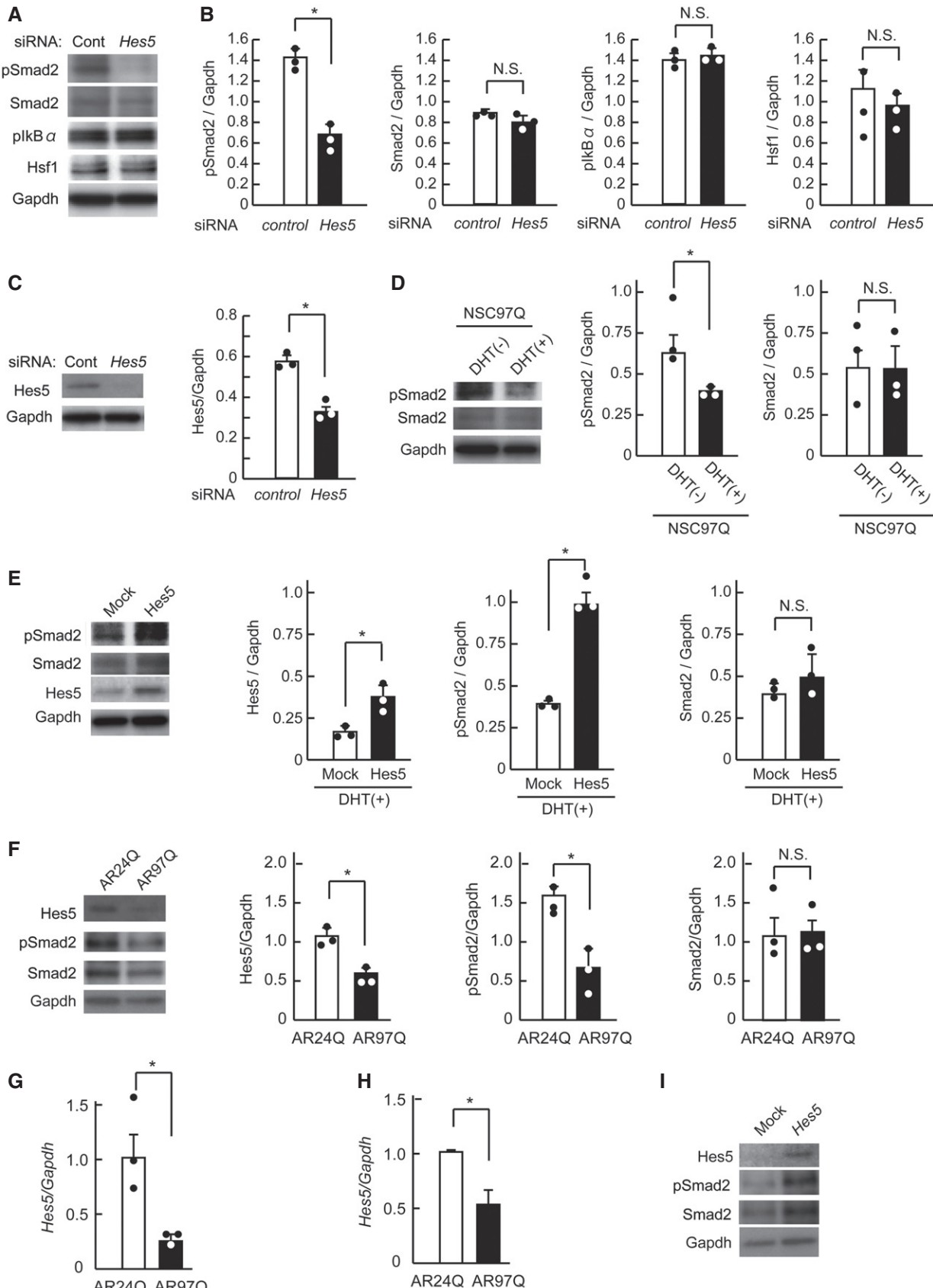

Figure 8.

◄

**Figure 8.   Hes5 plays a protective role against SBMA pathomechanism through Smad2 activation.**

A    Immunoblots of NSC34 cells treated with *Hes5* siRNA.
B    Quantification of protein levels of pSmad2, Smad2, phospho-IkBα, and heat shock factor-1 (Hsf1) (*n* = 3).
C    siRNA-mediated knockdown of *Hes5* down-regulated Hes5 protein level (*n* = 3).
D    Western blots of Smad2 and pSmad2 in NSC97Q cells with or without DHT treatment (*n* = 3).
E    Protein levels of Smad2, pSmad2, and Hes5 in NSC97Q cells transfected with *Hes5* vector (*n* = 3).
F    Immunoblots of Smad2, pSmad2, and Hes5 in primary cortical neurons treated with lentiviral vector containing AR-24Q and AR-97Q (*n* = 3).
G    Relative mRNA levels of *Hes5* in primary cortical neurons expressing AR-24Q and AR-97Q (*n* = 3).
H    Relative mRNA levels of *Hes5* in primary motor neurons expressing AR-24Q and AR-97Q (*n* = 6).
I     Western blots of Smad2, pSmad2, and Hes5 in primary cortical neurons expressing AR-97Q treated with mock plasmid or *Hes5*-containing lentiviral vector.

Data information: Unpaired *t*-test. Error bars, s.e.m. **P* < 0.05. N.S.: not significant. The exact *P*-value is in Appendix Table S3.
Source data are available online for this figure.

phosphorylation in both NSC34 cells and primary cortical neurons expressing AR-97Q. These findings suggest that Hes5 protects neurons from the toxicity of polyglutamine-expanded AR via activation of the Smad pathway, which has been shown to facilitate neuronal function and survival (Katsuno *et al*, 2010b).

Notably, neither RG108 therapy nor the genetic over-expression of *Hes5* can suppress pathogenic AR accumulation *in vivo* or *in vitro*, suggesting that DNMT inhibition suppresses neurodegeneration in SBMA without modulating the protein metabolism of pathogenic AR. Hence, RG108 could be used in combination with other therapies, such as androgen modifying drugs and heat shock protein inducers (Katsuno *et al*, 2010a,b; Kondo *et al*, 2013; Rusmini *et al*, 2016).

Recent studies have indicated that SBMA is not only a neurodegenerative disease but also a neuromuscular system disorder (Fischbeck, 2016; Giorgetti & Lieberman, 2016). Several lines of evidence indicate that abnormal mutant AR protein expression causes both neurogenic and myogenic pathologies (Cortes *et al*, 2014; Lieberman *et al*, 2014; Iida *et al*, 2015; Sahashi *et al*, 2015) in SBMA. However, we found that Dnmt1 level was substantially upregulated in only spinal motor neurons and not skeletal muscles (Fig 1G and H, and Appendix Fig S1), possibly because DNA methylation regulation varies with cell type. Namely, DNA methylation is a physiological mechanism for enabling cell-specific functions by controlling the gene expression patterns in each cell (Smith & Meissner, 2013), thus providing a possible molecular basis for the differential alteration of DNA methylation in the neurons and muscles of SBMA mice. In support of this notion, we previously documented cell-specific reactions against the pathogenic AR protein in the AR-97Q mouse model of SBMA (Kondo *et al*, 2013).

In summary, this study showed that DNA hyper-methylation by Dnmt1 is implicated in the pathogenesis of SBMA and is a novel epigenetic therapeutic target for SBMA. In addition, we identified that Hes5 is a target of RG108 and is a DNMT inhibitor; moreover, Hes5 plays a role as a robust neuroprotective molecule against polyglutamine-expanded AR toxicity. Thus, RG108 and its target gene *Hes5* are therapeutic candidates for SBMA.

# Materials and Methods

## Animals

We generated AR-97Q and AR-24Q mice by using the pCAGGS vector and maintained on a C57BL/6J background as previously mentioned (Katsuno *et al*, 2002, 2003). The mice were genotyped by PCR on tail DNA (Katsuno *et al*, 2002), and we randomly allocated them to each treatment group. We administered RG108 to the mouse ventricular at a concentration of 0.5 or 2.0 mg/dl in saline from 6 weeks old for 2 weeks using osmotic pump, which provides continuous infusion of agent for 2 weeks, and examined behavioral, biochemical, and pathological findings of the treated mice. As for the control group, we administered 20% of DMSO in saline. We used the littermates for phenotypic analysis to align the conditions as far as possible. We used approximately 90 male mice in total throughout this study including C57BL/6J as control. The number of mice used in each experiment is described in each figure legend. The mice had free access to water and a standard diet and were maintained on a 12-h light/dark cycle. We blindly assessed rotarod performance using an Economex Rotarod (Ugo Basile, Comerio, Italy) as described previously (Minamiyama *et al*, 2004). To measure grip strength, the same examiner (N.K.) placed the mice on wire netting with a Grip Strength Meter (MK-380M; Muromachi Kikai, Tokyo, Japan) and pulled them. All of the tests were performed weekly, and the data were analyzed retrospectively. All of the animal experiments were performed in accordance with the National Institutes of Health Guide for the Care and Use of Laboratory Animals and under the approval of the Nagoya University Animal Experiment Committee.

## Injection procedures

We stereotaxically injected RG108 into the right cerebral ventricle of 6-week-old AR-97Q mice anesthetized with mixed solution using an Alzet minipump 2002 (Alzet, Palo Alto, CA, USA) and a microinjection cannula (Eicom, Kyoto, Japan). The stereotaxic injection was performed as follows: 1.0 mm rostral to bregma, 1.5 mm lateral to midline, and 1.8 mm ventral from the dural surface. The same number of mice received the infusion of saline as control.

## Autopsy specimens

Autopsy specimens of the spinal cord were obtained from three genetically confirmed SBMA patients (52, 77, and 78 years old) and three control subjects (58, 64, and 70 years old) who had multiple systemic atrophy, Guillain–Barre syndrome, or neuropathy. Informed consent was obtained from relatives of people who we used autopsy material from, and the study was approved by the Ethics Committee of Nagoya University Graduate School of Medicine. The experiments conformed to the principles set out in the WMA Declaration of Helsinki, the Department of Health and Human

Services Belmont Report, and the Ethical Guidelines for Medical and Health Research Involving Human Subjects endorsed by the Japanese government. The collection of tissues and their use for this study conformed to the Ethics Guidelines for Human Genome/Gene Analysis Research and the Ethical Guidelines for Medical and Health Research Involving Human Subjects endorsed by the Japanese government and were approved by the Ethics Committee of Nagoya University Graduate School of Medicine.

## Immunoblotting

We homogenized mouse tissues in buffer containing 50 mM Tris–HCl (pH 8.0), 150 mM NaCl, 1% Nonidet P-40, 0.5% deoxycholate, 0.1% SDS, and 1 mM 2-mercaptoethanol with Halt Protease Inhibitor Cocktail (Thermo Fisher Scientific, NJ, USA) and then centrifuged them at $2,500 \times g$ for 15 min. Equal amounts of protein were separated by 5–20% SDS–PAGE and transferred to Hybond-P membranes (GE Healthcare, Piscataway, NJ, USA). The primary antibodies and their dilutions were as follows: AR (N20, 1:1,000; Santa Cruz Biotechnology, Santa Cruz, CA, USA), Dnmt1 (1:1,000; Abcam, Cambridge, MA, USA), Dnmt3a (1:1,000; Abcam), Dnmt3b (1:1,000; Abcam), Hes5 (1:1,000; Santa Cruz), and ChAT (1:1,000; Millipore, Billerica, MA, USA). Primary antibody binding was probed with horseradish peroxidase-conjugated secondary antibodies at a dilution of 1:5,000, and bands were detected by using an immunoreaction enhancing solution (Can Get Signal; Toyobo, Osaka, Japan) and enhanced chemiluminescence (ECL Prime; GE Healthcare). An LAS-3000 imaging system (Fujifilm, Tokyo, Japan) was used to produce digital images. The signal intensities of these independent blots were quantified using IMAGE GAUGE software version 4.22 (Fuji) and expressed in arbitrary units ($n = 3$ for each group). The membranes were reprobed, or the same samples were examined with an anti-GAPDH (1:5,000; Santa Cruz) antibody for normalization.

## Histology and immunohistochemistry

Mice anesthetized with a mixed solution that included medetomidine (0.3 mg/kg), midazolam (4 mg/kg), and butorphanol (5 mg/kg) were perfused with a 4% paraformaldehyde fixative in phosphate buffer (pH 7.4). The tissues were dissected, post-fixed in 10% phosphate-buffered formalin, and processed for paraffin embedding. The sections to be stained were treated with an anti-polyglutamine antibody (1C2) with formic acid for 5 min at room temperature; those to be incubated with anti-Dnmt1, anti-Dnmt3a, anti-Dnmt3b, anti-Hes5, or anti-ChAT antibody were boiled in 10 mM citrate buffer for 15 min. The primary antibodies and their dilutions for mouse and human specimens were as follows: polyglutamine (1:20,000; Millipore), Dnmt1 (1:1,000; Abcam), Dnmt3a (1:1,000; Abcam), Dnmt3b (1:1,000; Abcam), Hes5 (1:1,000; Santa Cruz), and ChAT (1:1,000; Millipore). We probed primary antibody binding with a polymer-labeled secondary antibody as part of the Envision+ system containing horseradish peroxidase (DakoCytomation, Glostrup, Denmark). For quantification of 1C2-positive cell rate and motor neuron size stained with Chat, five animals from each group were analyzed using ImageJ. As described previously (Kondo *et al*, 2013), we prepared at least 100 consecutive axial sections of the thoracic spinal cord and immunostained every tenth section with

an anti-polyglutamine 1C2 antibody. We counted the number of 1C2-positive cells in all of the motor neurons within the anterior horn of the ten axial sections from the thoracic spinal cord and more than 50 neurons in five randomly selected × 400 microscopic fields of the ten sections of each group samples under a light microscope (Bx51; Olympus, Tokyo, Japan). For counting, motor neurons in a given 6-μm-thick section were defined by their presence within an anterior horn and an obvious nucleolus. For the purposes of calculating the cell size of motor neurons in the spinal anterior horn with Chat staining, more than 50 neurons in randomly selected areas were examined. We also measured the immunoreactivity of Dnmt1, Dnmt3a, Dnmt3b, Hes5, and 5mC in more than 20 neurons in 3–5 nonconsecutive sections, and three mice from each group were analyzed. Intensities were measured with ImageJ.

## DNA methylation array analysis

Genomic DNA from human neuroblastoma SH-SY5Y cells stably expressing human AR containing 24 or 97 glutamine residues (AR-24Q or AR-97Q) was isolated from cell pellets using a DNA isolation kit for cells and tissues (Roche, Mannheim, Germany) according to the manufacturer's instructions. DNA concentrations were measured by spectrophotometry (NanoDrop ND-1000 Spectrometer, Thermo Fisher Scientific, Wilmington, DE, USA), and DNA quality was estimated with the A260/A280 absorbance ratio. Array-based specific DNA methylation analysis was performed using Human Methylation 450K BeadChip technology (Illumina, CA, USA). In this study, six samples were analyzed for more than 450,000 CpG sites at single nucleotide resolution with 96% coverage of CpG islands and 99% coverage of the RefSeq gene. Probes were distributed in CpG island shelves, CpG island shores, CpG islands, 5′ UTR promoter regions, first exons, gene bodies, and 3′ UTRs. The GenomeStudio Methylation Software Module (Illumina) was used to perform these analyses and assess image intensities.

## siRNA

We transfected NSC34 cells stably expressing AR-97Q with a 10 nM siRNA oligonucleotide using Lipofectamine 3000 (Invitrogen, MD, USA) according to the manufacturer's instructions. NSC34 cells stably expressing AR-97Q were transfected with the following oligonucleotides to knock down *Dnmt1, Dnmt3a, Dnmt3b*, and *Hes5*: *Dnmt1*, MISSION siRNA ID: SASI_Mm01_00024007; *Dnmt3a*, MISSION siRNA ID: SASI_Mm01_00201687; *Dnmt3b,* MISSION siRNA ID: SASI_Mm02_00286412; and *Hes5*, MISSION siRNA ID: SASI_Mm01_00020733 (all from Sigma-Aldrich, USA). We used MISSION siRNA Universal Negative Control (Sigma-Aldrich) as the control for *Dnmt1, Dnmt3a, Dnmt3b*, and *Hes5* siRNA.

## Transfection

Cells were transfected with either a *Hes5* expression vector (gifted by Ryuichiro Kageyama) or a mock vector using Opti-MEM (Invitrogen) and Lipofectamine 3000 (Invitrogen) and then differentiated in DMEM supplemented with 10 nM 5α-dihydrotestosterone. The mouse *Hes5* vector and mock vector have EF promoter and

HA tag. FCS (10%) was added to the medium for the cell viability assay, but no serum was used for the cell toxicity assay because of the LDH measurement properties. Cells were transfected with the *Hes5* or mock vector on day 0 for the cell viability and LDH assays.

## Cell viability assays

The WST-8 (Roche) and LDH cell viability assays were performed using a cytotoxicity LDH assay kit (Dojindo) according to the manufacturer's instructions. The cells were cultured in 6-well plates, and after each treatment, they were incubated with the WST-8 substrate for 2 h and then spectrophotometrically assayed at 450 nm using a plate reader (Powerscan HT; Dainippon Pharmaceutical, Tokyo, Japan). For the toxicity assays, NSC34 cells stably expressing AR-97Q were differentiated the day after being plated in DMEM with the same supplement used after transfection. RG108 at concentrations of 0.1, 1, and 10 μM was administered to NSC34 cells stably expressing AR-97Q in a serum-free medium for 3 days after differentiation. Two hours after treatment, we performed the cell viability assay. The cell viability and cytotoxicity assays were performed 24 h after drug administration.

## Quantitative real-time RT–qPCR

The mRNA levels of *CDC25B, GFRA3, NPY, HES5, SCTR, LEF1-AS1,* and *CABS1* were analyzed by real-time RT–PCR as described previously (Miyazaki *et al*, 2012). Briefly, total RNA was isolated using an RNeasy Mini Kit (Qiagen, Hilden, Germany) from cell pellets and spinal cords by homogenization in TRIzol Reagent (Invitrogen) according to the manufacturer's instructions. Total RNA was reverse-transcribed using ReverTra Ace (Toyobo). PCR primers were designed to amplify human *CDC25B* (5′-CACTCGGTCC CAGTTTTGTT-3′ and 5′-GTTTGGGTATGCAAGGCACT-3′), human *GFRA3* (5′-GGAACTTGTGCAACAGAGCA-3′ and 5′-ACCCTTCCAG CATTTCACAC-3′), human *NPY* (5′-CCTCATCACCAGGCAGAGAT-3′ and 5′-TAGGAAAAGGCCAGAGAGCA-3′), human *HES5* (5′-GCCC GGGGTTCTATGTATT-3′ and 5′-GAGTTCGGCCTTCACAAAAG-3′), human *SCTR* (5′-GTGTCCTTCATCCTTCGTGC-3′ and 5′-AGTGTGT GAAGGTAGAGGCC-3′), human *LEF1-AS1* (5′-ACAGATCACCCCACC TCTTG-3′ and 5′-GAGGCTTCACGTGCATTAGG-3′), human *CABS1* (5′-ACCTCTGAAGTCTCTGGCAC-3′ and 5′-TTTCTTCAGGAGCAG GAGGG-3′), and mouse *Hes5* (5′-CTTCTGCGAAGTTCCTGGTC-3′ and 5′-ATGTGGACCTTGAGGTGAGG-3′). Real-time RT–PCR was carried out in a total volume of 50 μl, comprising 25 μl of 2 × QuantiTect SYBR Green PCR Master Mix (Qiagen, CA, USA) and 10 μM of each primer. PCR products were detected using the iCycler system (Bio-Rad, Hercules, CA, USA). The reaction conditions were as follows: 95°C for 15 min, followed by 45 cycles of 15 s at 94°C, 30 s at 55°C, and 30 s at 72°C. As internal standard controls, mouse *beta-2 microglobulin* expression was simultaneously quantified using the primers 5′-AAGCCGAACATACTGAACTGC-3′ and 5′-GTGTGAGC CAGGATATAGAAAGAC-3′, and human *beta-2 microglobulin* was simultaneously quantified using the primers 5′-TTTCATCCATC CATCCGACATTGA-3′ and 5′-CCTCCATGATGCTGCTTACA-3′. The relative strength indices were computed as the signal intensity of each sample divided by that of *beta-2 microglobulin* in each of the cell lines.

## Methylation-specific PCR

Genomic DNA was bisulfite-converted with an EZ DNA Methylation-Gold Kit (Zymo Research Corporation, USA) according to the manufacturer's instructions. Methylation-specific PCR was performed using human *HES5* and mouse *Hes5* primer sets for both methylated and unmethylated DNA. MSP analysis was conducted on NSC34 and SH-SY5Y cells using an EpiScope MSP kit (TaKaRa) according to the manufacturer's protocol. Episcope unmethylated HCT116DKO gDNA (TaKaRa) and Episcope methylated HCT116 gDNA (TaKaRa) were used as the negative and positive controls, respectively. Negative and positive mouse methylated and unmethylated DNA controls were prepared as described previously (Suzuki *et al*, 2013). The human methylated (*HES5*-M) and unmethylated (*HES5*-UM) primer sequences were as follows: 5′-GGTTAGTTT GGGGTTAGAGTTTTTC-3′ and 5′-AAAAACGAACCTACTCCTCTCG-3′ for *HES5*-M, and 5′-GTTAGTTTGGGGTTAGAGTTTTTG-3′ and 5′-AAAAACAAACCTACTCCTCTCACC-3′ for *HES5*-UM. The mouse methylated (*Hes5*-M) and unmethylated (*Hes5*-UM) primer sequences were as follows: 5′-GTTTAGTTTTAAGGAGAAAAA TCGA-3′ and 5′-TACTACTATTAATACGATCCCGACG-3′ for *Hes5*-M, and 5′-GTTTAGTTTTAAGGAGAAAAATTGA-3′ and 5′-CTACTAT TAATACAATCCCAACACA-3′ for *Hes5*-UM. The PCR conditions were as follows: initial denaturation at 95°C for 30 s, followed by 40 cycles of denaturation at 98°C for 5 s, annealing at 60°C for 30 s, and extension at 72°C for 1 min. Agarose gel electrophoresis and ethidium bromide staining were then performed. DNA was visualized using a UV transilluminator, and gels were photographed with a digital camera.

## Lentiviral production

A lentiviral vector construct containing mouse *Hes5* was produced by TaKaRa bio (TaKaRa). Lentivirus was prepared with HEK293T cells by transfection using Lentiviral High Titer Packaging Mix and TransIT-293 Transfection Reagent (TaKaRa). The lentivirus-containing supernatant was collected 48 h after transfection. The viral titer was measured using a Lenti-X qRT-PCR Titration Kit (TaKaRa).

## Primary neuron culture

We obtained the primary cortical neurons by dissecting C57BL/6J mouse embryos at E16. To prepare the primary motor neurons, we dissected the spinal cord from C57BL/6J mouse embryos at E13 as described previously (Minamiyama *et al*, 2012). Shortly, after removing the meninges, dorsal root ganglia, and the dorsal half of the spinal cord, we separated the motor neurons by using Sumitomo dissociation solution (Sumitomo Bakelite). We plated the cells at a density of $2 \times 10^5$ cells in a 24-well culture plate with Sumitomo nerve-culture medium (Sumitomo Bakelite). On day 3, we infected the neurons with approximately $1.0 \times 10^7$ IFU/ml of lentivirus expressing truncated human AR with 24Q and 97Q. Moreover, lentivirus expressing murine Hes5 was infected at a range around $1.0 \times 10^7$ IFU/ml. Three hours after infection, we removed the virus medium. We then cultured the neurons for two additional days and harvested them on day 6, followed by protein and mRNA extraction.

**The paper explained**

**Problem**

Transcriptional dysregulation in the susceptive neurons within the central nervous system is a key mechanism of polyglutamine-related diseases including spinal and bulbar muscular atrophy (SBMA), a hereditary neuromuscular disease caused by CAG trinucleotide repeat expansion in the gene encoding androgen receptor. Histone deacetylation and DNA methylation are the major epigenetic modifications which silence gene expression, and thus are possible therapeutic targets of polyglutamine-inducing diseases. However, compared with histone deacetylation, there is little investigation regarding DNA methylation.

**Result**

We revealed that DNA methyltransferase 1 (Dnmt1) is expressed at a high level in the spinal motor neurons of model mice and patients with SBMA. RG108, a DNA methyltransferase (Dnmt) inhibitor, suppressed cell death of a cellular model of SBMA and improved the motor functions and survival of SBMA model mice, without inhibiting nuclear accumulation of pathogenic androgen receptor. We identified Hes5 as a target of RG108 by performing DNA methylation array analysis and confirmed that this compound restores the transcription of *Hes5* through DNA demethylation of its promotor region. Furthermore, we found that over-expression of *Hes5* leads to up-regulation of pSmad2 in SBMA model cells, corresponding to our previous finding that pSmad2 has a protective role in the neurodegenerative process of SBMA.

**Impact**

This study indicated that abnormal DNA hyper-methylation underlies the neurodegeneration in SBMA and DNA methyltransferase inhibitor might be a novel therapeutic candidate for SBMA.

## Statistical analysis

Based on previous studies, we estimated that a sample size of 20 mice per group would provide 80% power to detect 0.33 hazard ratio for survival with 30-week observation between treatment groups (log-rank test), with a two-sided $\alpha$ level of 0.05. We randomly allocated the mice to each treatment group, and the investigator blindly performed the behavior tests. The data were analyzed using unpaired *t*-tests for two-group comparisons, ANOVA with Tukey's *post hoc* tests for multiple comparisons, and Kaplan–Meier and log-rank tests for survival rate using IBM SPSS Statistics version 24 for Windows (IBM) and Excel, and denoted *P*-values of 0.05 or less were considered statistically significant.

# Data availability

The DNA methylation array analysis data from this publication have been deposited to the Gene Expression Omnibus database (https://www.ncbi.nlm.nih.gov/geo/) and assigned the identifier GSE113308.

**Expanded View** for this article is available online.

## Acknowledgements

We thank Dr. Ryuichiro Kageyama for kindly providing Hes5 vector. This work was supported by KAKENHI grants from MEXT/JSPS, Japan (Nos. 17H04195 and 16K15480); grants from Japan Agency for Medical Research and Development (Nos. 17ek0109221h0001, 16dk0207026h0001, and 15ek0109165); and a grant from The Naito Foundation. There were no other funding sources, and the investigators had sole discretion over study design; collection, analysis, and interpretation of data; writing of the report; and the decision to submit it for publication.

## Author contributions

Project planning was performed by NK and MKats; experimental work by NK, GT, KS, MI, MKata, HN, YT, HA, HK, and KS; data analysis by NK, GT, and MKats; statistical analysis by NK, GT, and AH; composition of the first draft of the manuscript by NK and MKats; and manuscript layout by NK, YK, GS, and MKats.

## Conflict of interest

The authors declare that they have no conflict of interest.

## For more information

Department of Neurology, Nagoya University Graduate School of Medicine: https://www.med.nagoya-u.ac.jp/neurology/en/index.html.

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
