## [Review Process File · EMBO Molecular Medicine]

DNA methylation inhibitor attenuates polyglutamine-induced neurodegeneration by regulating Hes5

Naohide Kondo, Genki Tohnai, Kentaro Sahashi, Madoka Iida, Mayumi Kataoka, Hideaki Nakatsuji, Yutaka Tsutsumi, Atsushi Hashizume, Hiroaki Adachi, Haruki Koike, Keiko Shinjo, Yutaka Kondo, Gen Sobue, Masahisa Katsuno

Review timeline:

Submission date:	1 October 2017
Editorial Decision:	13 November 2017
Revision received:	27 February 2018
Editorial Decision:	6 April 2018
Revision received:	4 May 2018
Extra communication:	17 May 2018
Editorial Decision:	12 June 2018
Revision received:	7 January 2019
Editorial Decision:	11 January 2019
Revision received:	1 March 2019
Accepted:	1 March 2019

Editor: Céline Carret

Transaction Report:

1st Editorial Decision

13 November 2017

Thank you for the submission of your manuscript to EMBO Molecular Medicine. We have now heard back from the two referees whom we asked to evaluate your manuscript.

As you will see from the reports below, the referees find the data intriguing. Unfortunately, at this stage the work needs to be considerably strengthened to be more conclusive, and both referees provide good suggestions for that. Additional controls and experiments and adding mechanistic understanding seem to be required for the paper to be further evaluated in EMBO Molecular Medicine.

Given that the referees find the message interesting, we would be willing to consider a revised manuscript with the understanding that the referee concerns must be fully addressed and that acceptance of the manuscript would entail a second round of review. I should remind you that EMBO Molecular Medicine encourages a single round of revision only and therefore, acceptance or rejection of the manuscript will depend on the completeness of your responses included in the next, final version of the manuscript. I realize that addressing the referees' comments in full would involve a lot of additional experimental work and resources and I am uncertain whether you will be able (or willing) to return a revised manuscript within the 3 months deadline. I would also understand your decision if you chose to rather seek rapid publication elsewhere at this stage.

I look forward to seeing a revised form of your manuscript as soon as possible.

Should you find that the requested revisions are not feasible within the constraints outlined here and choose, therefore, to submit your paper elsewhere, we would welcome a message to this effect.

***** Reviewer's comments *****

Referee #1 (Comments on Novelty/Model System for Author):

Better control should be used as detailed to Authors

Referee #1 (Remarks for Author):

The study by Kondo *al coll.* Reports that the toxicity of the mutant AR responsible for SBMA could be ascribed to altered DNA methylation, and that this alteration could be reverted by the inhibition of this mechanisms using DNA methylation inhibitor such as RG108. In addition, the Authors have demonstrated that the hyper-methylation of the Hes5 gene correlates with the AR-97Q toxicity and that overexpression of Hes5 rescues from SBMA phenotype in cell models of the disease. While the study is potentially interesting, several points must be addressed in order to definitely prove that DNA methylation is relevant in SBMA.

Major points:

1. Results, page 5, lines 13-14 and Figure 1. It is clear that the levels of the Dnmt1 protein, measured in WB or determined qualitatively in IF are increased. It will be of interest to determine whether this increase is due to an enhanced transcription in spinal cord or rather to a decreased turn over, which could be associated to the impact of the AR-97Q on the protein quality control system. In fact, AR-97Q may alter proteasome functions and this could reduce the overall clearance of a series of proteins processed via the proteasome. A RT-qPCR could be performed on RNA samples derived from spinal cord of wt and AR-97Q mice.

2. Why the Authors decided to use the littermate non-transgenic mice, instead of the tg SBMA mice expressing the wt AR which have been developed by the same group in 2002? To exclude that the alterations observed (at least in basal conditions for the expression of Dnmts in Fig 1 and Hes5 in Fig 5H,I) are due to the presence of the overexpressed wtAR in motoneurons instead of the mutant AR-97Q, samples derived from the tg AR overexpressing mice must be included in these analyses.

3. Page 6, line 20: The term "expression" is not correct. In WB (Figure 2 panels A and B), protein "levels" are detected. Changes in these levels may be associated to enhanced gene expression (measured determining RNA levels) and/or reduced degradation. As in point 1, a quantitative real time-PCR analysis is required to prove that enhanced Dnmt1 protein levels are linked to the transcriptional activation of the DNMT1 gene in response to the presence of testosterone activated AR-97Q in NSC34 cells. In addition, please modify as follow: "...and Dnmt3b was not changed (Fig. 2A and B); the same phenomena were observed in spinal cord lysates of SBMA mice, as determined by western blot (see Fig. 1A and B)..." In fact, in page 6, line 23 the WB mentioned refers to spinal cord (which includes motoneurons along with many other cell types, e.g. astrocytes other neurons, etc.) and this analysis is reported in Fig. 1A and B. Fig. 2A and B mentioned here is related to the WB in NSC34 cells. In all the experiments reported in Figure 2, it is required to use the control cells using NSC34 cells not treated with androgens (unactivated AR); in fact, these are stable transfected cells and the differences between NSC24Q and NCS97Q could be due to their clonal selection. The direct comparison between untreated and DHT treated cells will prove whether Dnmts levels are dependent on the AR-97Q toxicity triggered by androgens. No changes in Dnmt1 should be present in NSC97Q cells not exposed to DHT.

4. Page 6, lines 24-24, Page 7, line 1 and Figure 2E: Is there an effect of Dnmt1 (and the other Dnmts) downregulation on the cell viability of testosterone treated NSC24Q? These data should be added into the supplementary material. The same is true for the experiments with RG108; the data obtained on wt cells (NSC24Q) should be included in the manuscript

5. Page 8, Hes5 silencing in SH-SY5Y cells. Why all these experiments were conducted in neuroblastoma cells and the experiments reported in Fig 2 in NSC34 cells?

6. Page 7, line 13: Why pre-symptomatic SBMA mice have been selected to start the treatment with RG108

7. Page 8, lines 20-23. Please re-phrase this paragraph, which is not clear to readers.

8. Page 9, line 16: Why Hes5 is diffused in the cytoplasm and nuclei of motoneurons of wt mice and it is only cytoplasmic in motoneurons of SBMA mice?

9. Figures 5A, 5D, 5G, 5E, 6B, 6D, 7A-H, as in the case of Fig 2 it is required to add as control the untreated NSC34 or SH-SY5Y stably expressing AR-97Q. Also in these cases the cells used are stably transfected cells and the differences observed may derive from clonal selection between AR-24Q and AR-97Q expressing cells. The analysis in cells not exposed to androgens in comparison the cells treated with DHT will allow to ascribe the variations of the levels of the proteins considered directly to the AR-97Q toxicity triggered by androgens. No changes in their levels should be noted in cells not exposed to DHT in all condition tested.

Minor Points

1. Page 9, line 3: the analysis performed is a quantitative analysis. RT-PCR usually refers to the qualitative PCR analysis performed on retro-transcribed RNA. It will be better to use RT-qPCR or Real Time-PCR.

2. Please used the term "Wild type" instead of "Wild" in all figures in which these animals have been used.

Referee #2 (Comments on Novelty/Model System for Author):

See comments to authors for 1 and 4. One issue is that validation in the NSC transformed cell line model is not sufficient.

Referee #2 (Remarks for Author):

In the Kondo et al. paper, a role for altered methylation in the polyglutamine disease SBMA is sought. The authors show Dnmt1 is upregulated within spinal motor neurons and suppression by siRNA or by a chemical inhibitor RG108 can suppress neuronal cell death. The authors translated the inhibitor into a pre-clinical mouse model and found evidence for suppression of SBMA neurodegenerative phenotypes. Intriguingly, this occurred downstream of polyglutamine aggregation, as aggregation was not affected. The authors show that cell viability in a NSC cell model of SBMA enabled by RG108 treatment depends on Hes5 expression, and that re-expression of Hes5 is sufficient to rescue viability.

While the topic of methylation in neurodegeneration has not been addressed much, making the work intriguing, there are a number of problems with the investigation that undermine one's confidence in the results. The authors' explanation for which genes are affected by altered methylation is very superficial, and what Hes5 is doing is entirely unclear. As the validation of Hes5 takes place in only the transformed cell line model of SBMA, it seems rather premature to implicate Hes5 without providing a more thorough understanding of what it is doing. More concrete data is needed to balance what appears at times as speculation.

Specific concerns that should be addressed to strengthen the manuscript are as follows:

- 1) The AR97 mouse model shows over-expression of AR from an out-of-context promoter with little relationship to the normal regulation of AR. The authors need to exclude an effect upon AR transgene expression by RG108 as its mechanism of beneficial action.
- 2) A number of claims made by the authors are only supported by sample images without quantitation. All of these findings need to be rigorously quantified and presented with statistical analysis. Also, the number of biological samples and number of frames used for image quantitation should be included in the Figure legends so that readers (and reviewers) can determine if the conclusions are valid. The problematic Figures are as follows:
 - Fig 1C
 - Fig 1D
 - Fig 1E,

- Fig 1F
- Fig 2D
- Fig 3H
- Fig 5H
- Fig 6F
- Fig S1
- Fig S2
- Fig S3
- Fig S4
- Fig S5

3) All the cell model experiments use DHT+ conditions. Is the over-expression of Dnmt1 or suppression of Hes5 dependent on polyglutamine AR alone or must agonist treatment be included? These variations should be included as controls in such studies.

4) As noted above, the Hes5 connection to SBMA disease is tenuous at best, as the validation work is done in the NSC transformed cell line model, and the authors appear to have no idea whatsoever as the action of Hes5. Going further would make the work more convincing.

Minor points:

- There remain quite a few grammatical errors and typos, the authors should review the manuscript for these
- The control for no over-expression of Dnmt1 in muscle tissue in Fig. 1G,H (and S1-3) is welcome, however the authors suggest localization of Dnmt1 may be most important. Can the authors provide any quantified evidence, such as subcellular localization or 5-methylcytosine levels, in muscle tissue?
- Can the authors kindly provide the rationale for choosing RG108?
- Clinical inhibitors for DNMTs, including azacytidine and decitabine, have relatively high toxicity. Was the authors' hope with RG108 to more specifically target the specific DNMT affected? Can the authors provide any evidence of toxicity to blood counts in mice, as is found in patients? If these data are not available, the authors should qualify their statement suggesting DNMT inhibitors may be useful in humans; mice are in a sterile environment and DNMTs are typically administered for short periods of time.
- Figures 3A-D are missing significance labels
- Please document the backbone and other characteristics (promoter?) of the Hes5 expression vector and mock vector.

1st Revision - authors' response

27 February 2018

Referee #1:

Major points:

Point 1. Results, page 5, lines 13-14 and Figure 1. It is clear that the levels of the Dnmt1 protein, measured in WB or determined qualitatively in IF are increased. It will be of interest to determine whether this increase is due to an enhanced transcription in spinal cord or rather to a decreased turn over, which could be associated to the impact of the AR-97Q on the protein quality control system. In fact, AR-97Q may alter proteasome functions and this could reduce the overall clearance of a series of proteins processed via the proteasome. A RT-qPCR could be performed on RNA samples derived from spinal cord of wt and AR-97Q mice.

Response 1: We thank the Referee for this important suggestion. In response to the proposal, we performed RT-qPCR of *Dnmt1*, *Dnmt3a*, and *Dnmt3b* using spinal cord samples from each group of mice (Fig. 1C). This analysis revealed that *Dnmt1* mRNA level was up-regulated in AR-97Q mice as well as protein level.

Point 2. Why the Authors decided to use the littermate non-transgenic mice, instead of the tg SBMA mice expressing the wt AR which have been developed by the same group in 2002? To exclude that the alterations observed (at least in basal conditions for the expression of Dnmts in Fig 1 and Hes5 in Fig 5H,I) are due to the presence of the overexpressed wtAR in motoneurons instead of the mutant AR-97Q, samples derived from the tg AR overexpressing mice must be included in these analyses.

Response 2: In response to the Referee's comment, we added the data of AR-24Q mice in Fig. 1A, 1B, 1C, 1D, 1E, 5I, and 5J, so that we excluded the influence of normal AR to the protein level and mRNA level of Dnmt1, Dnmt3a, Dnmt3b, and Hes5.

Point 3-1. Page 6, line 20: The term "expression" is not correct. In WB (Figure 2 panels A and B), protein "levels" are detected. Changes in these levels may be associated to enhanced gene expression (measured determining RNA levels) and/or reduced degradation. As in point 1, a quantitative real time-PCR analysis is required to prove that enhanced Dnmt1 protein levels are linked to the transcriptional activation of the DNMT1 gene in response to the presence of testosterone activated AR-97Q in NSC34 cells.

Response 3-1: We changed the term "expression" to "protein level" or "level" for the part of manuscript describing western blotting. In order to evaluate the gene expression level of *Dnmt1*, we performed RT-qPCR of NSC24Q and NSC97Q cells, and confirmed that *Dnmt1* mRNA is expressed at a higher level in NSC97Q compared with NSC24Q (Fig. 2B).

Point 3-2. In addition, please modify as follow: "...and Dnmt3b was not changed (Fig. 2A and B); the same phenomena were observed in spinal cord lysates of SBMA mice, as determined by western blot (see Fig. 1A and B)..." In fact, in page 6, line 23 the WB mentioned refers to spinal cord (which includes motor neurons along with many other cell types, e.g. astrocytes other neurons, etc.) and this analysis is reported in Fig. 1A and B. Fig. 2A and B mentioned here is related to the WB in NSC34 cells.

Response 3-2: We changed the sentence according to Referee's suggestion (Page 7, Line 1-2).

Point 3-3. In all the experiments reported in Figure 2, it is required to use the control cells using NSC34 cells not treated with androgens (unactivated AR); in fact, these are stable transfected cells and the differences between NSC24Q and NCS97Q could be due to their clonal selection. The direct comparison between untreated and DHT treated cells will prove whether Dnmts levels are dependent on the AR-97Q toxicity triggered by androgens. No changes in Dnmt1 should be present in NSC97Q cells not exposed to DHT.

Response 3-3: To address the comment, we added the data comparing protein levels of Dnmts and *Dnmt1* mRNA level in NSC97Q with or without DHT treatment (Fig. 2C and D). Moreover, we analyzed the influence of Dnmts knockdown on the cell viability of DHT-untreatd NSC97Q cells (Fig. 2F). Furthermore we evaluate the effect of RG108 to NSC97Q cells without DHT (Fig. 2I). The results of these additional experiments showed that Dnmt1 is not up-regulated in DHT-untreated NSC97Q cells, and that neither siRNA-mediated knockdown of *Dnmts* nor RG108 treatment alters the cell viability of NSC97Q cells without DHT treatment.

Point 4. Page 6, lines 24-24, Page 7, line 1 and Figure 2E: Is there an effect of Dnmt1 (and the other Dnmts) downregulation on the cell viability of testosterone treated NSC24Q? These data should be added into the supplementary material. The same is true for the experiments with RG108; the data obtained on wt cells (NSC24Q) should be included in the manuscript.

Response 4: According the Referee's suggestion, we added the data of WST-8 assay of DHT-treated NSC24Q cells with siRNA-mediated knockdown of Dnmts (Supplementary Fig. 9) and RG108 treatment to DHT-treated NSC24Q (Fig. 2H and J). Both knockdown of Dnmts and RG108 had no effect on the cell viability of NSC24Q with DHT treatment.

Point 5. Page 8, Hes5 silencing in SH-SY5Y cells. Why all these experiments were conducted in neuroblastoma cells and the experiments reported in Fig 2 in NSC34 cells?

Response 5: The DNA methylation array analysis we utilized was developed for human materials, but could not be applied to mouse samples at the time we performed experiments. Therefore we analyzed the samples of SH-SY5Y, human neuronal cells. In order to confirm that the phenomena observed in human cells are reproduced in murine cells, we performed all the experiments using cell samples derived from SH-SY5Y (Fig. 5 A, 5B, 5C, 5D, 5F, 5G, and 5H; Supplementary Fig. 12; Fig. 6A and 6B) and those from NSC34 (Fig. 2A and 2C; Supplementary Fig.13 and Fig. 6 C and 6D) in parallel.

Point 6. Page 7, line 13: Why pre-symptomatic SBMA mice have been selected to start the treatment with RG108.

Response 6: Our preliminary experiments revealed that Dnmt1 protein level was already elevated in the spinal cord of 6-week-old AR-97Q mouse (Figure for Referee shown below). This is the reason why we decided to start the brain injection treatment with RG108 to 6-week-old age mice.

Point 7. Page 8, lines 20-23. Please re-phrase this paragraph, which is not clear to readers.

Response 7: We changed the sentence to “DNA methylation array analysis using these cells revealed that DNA methylation of CpG islands is intensified in several genes. However, total DNA methylation level was not altered between SH24Q and SH97Q (Supplementary Fig. 11).” (Page 9, Line 7-9)

Point 8. Page 9, line 16: Why Hes5 is diffused in the cytoplasm and nuclei of motorneurons of wt mice and it is only cytoplasmic in motorneurons of SBMA mice?

Response 8: To address this issue, we performed western blot using nuclear and cytoplasmic fraction of spinal cord lysate obtained from wild-type and AR97Q mice. Both in wild-type and AR97Q mice, Hes5 showed cytoplasm-dominant localization (Figure for Referee shown below). Although we cannot exclude the possibility that nuclear localization of Hes5 is somehow impaired in AR97Q mice, the western blot indicates that Hes5 is down-regulated both in nucleus and cytoplasm in SBMA model mouse, presumably resulting in the faint nuclear staining in the immunohistochemistry of the AR97Q mice.

Point 9. Figures 5A, 5D, 5G, 5E, 6B, 6D, 7A-H, as in the case of Fig 2 it is required to add as control the untreated NSC34 or SH-SY5Y stably expressing AR-97Q. Also in these cases the cells used are stably transfected cells and the differences observed may derive from clonal selection between AR-24Q and AR-97Q expressing cells. The analysis in cells not exposed to androgens in comparison the cells treated with DHT will allow to ascribe the variations of the levels of the proteins considered directly to the AR-97Q toxicity triggered by androgens. No changes in their levels should be noted in cells not exposed to DHT in all condition tested.

Response 9: According to the comment, we added the data comparing DHT(-) and DHT(+) as shown in the Table for Referee shown below. As AR aggregation is not detectable in NSC97Q cell without DHT treatment, we did not perform the counterpart experiment for Fig. 7G and H.

The data Referee pointed	The corresponding data in the revise version
Fig. 5A	Fig. 5C
Fig. 5D (now Fig. 5F)	Fig. 5G
Fig. 5G (now Supplementary Fig. 13B)	Supplementary Fig. 13D
Fig. 5E (now Fig. 5H)	Supplementary Fig. 12
Fig6. B, D	Supplementary Fig. 14A, B
Fig. 7A-F	Supplementary Fig. 15A-F

Minor Points

Point 1. Page 9, line 3: the analysis performed is a quantitative analysis. RT-PCR usually refers to the qualitative PCR analysis performed on retro-transcribed RNA. It will be better to use RT-qPCR or Real Time-PCR.

Response: We are thankful to the suggestion. We changed RT-PCR to RT-qPCR throughout the manuscript.

Point 2. Please used the term "Wild type" instead of "Wild" in all figures in which these animals have been used.

Response: We changed the term from "Wild" to "Wild-type".

Referee #2

Point 1. The AR97 mouse model shows over-expression of AR from an out-of-context promoter with little relationship to the normal regulation of AR. The authors need to exclude an effect upon AR transgene expression by RG108 as its mechanism of beneficial action.

Response 1: We thank the Referee for pointing this important issue. We now confirmed that relative mRNA level of human *AR* was not altered by RG108 treatment (Fig. 4E). This data indicated that RG108 had does not suppress AR transgene expression.

Point 2. A number of claims made by the authors are only supported by sample images without quantitation. All of these findings need to rigorously quantified and presented with statistical analysis. Also, the number of biological samples and number of frames used for image quantitation should be included in the Figure legends so that readers (and Referees) can determine if the conclusions are valid. The problematic Figures are as follows:

- Fig 1C
- Fig 1D
- Fig 1E,
- Fig 1F
- Fig 2D
- Fig 3H
- Fig 5H
- Fig 6F
- Fig S1
- Fig S2
- Fig S3
- Fig S4
- Fig S5

Response 2: In response to the comment, we quantified the data as described below. Furthermore, we described the number of samples analyzed for quantitation in the figure legends and the Material and methods section (Page 17, Line 24 to Page 18 Line 3).

The data Referee pointed	Quantification of data in the revise version
Fig. 1C-E (now Fig. 1D)	Fig. 1E
Fig. 1F	Mentioned in the manuscript (Page 5, Line 23-24)
Fig. 2D (now Supplementary Fig. 7)	Supplementary Fig. 7
Fig. 3H	Fig. 3I
Fig. 5H (now Fig. 5I)	Fig. 5I
Fig. 6F	Fig. 6G
Supplementary Fig. 1	Supplementary Fig. 1
Supplementary Fig. 2	Supplementary Fig. 2
Supplementary Fig. 3	Supplementary Fig. 3
Supplementary Fig. 4	Supplementary Fig. 4
Supplementary Fig. 5 (now Supplementary Fig. 6)	Supplementary Fig. 6

Point 3. All the cell model experiments use DHT+ conditions. Is the over-expression of Dnmt1 or suppression of Hes5 dependent on polyglutamine AR alone or must agonist treatment be included? These variations should be included as controls in such studies.

Response 3: We added the data of cell model experiments without DHT treatment as shown in Table for Referee shown below.

The figure in the initial submission	The data including DHT-untreated cells in the revised version
Fig. 2A	Fig. 2C
Fig. 2C (now Fig. 2E)	Fig. 2F
Fig. 2E (now Fig. 2G)	Fig. 2I
Fig. 5A	Fig. 5C
Fig. 5D (now Fig. 5F)	Fig. 5G
Fig. 5G (now Supplementary Fig. 13B)	Supplementary Fig. 13D
Fig. 5E (now Fig. 5H)	Supplementary Fig. 12
Fig. 6B, D	Supplementary Fig. 14A, B
Fig. 7A-F	Supplementary Fig. 15A-F

Point 4. As noted above, the Hes5 connection to SBMA disease is tenuous at best, as the validation work is done in the NSC transformed cell line model, and the authors appear to have no idea whatsoever as the action of Hes5. Going further would make the work more convincing.

Response 4: In order to explore the role of Hes5 in SBMA, we compared the Hes5 protein and mRNA levels in primary cortical neurons expressing AR24Q and AR97Q using lentivirus infection. AR97Q with DHT reduced the levels of Hes5 both in western blot and RT-qPCR in the primary cortical neurons as shown in NSC97Q (Fig. 8F, G). Furthermore, we confirmed that Hes5 reduction was observed in DHT-treated primary motor neurons expressing AR97Q compared with those expressing AR24Q (Fig. 8H). To further clarify the molecular basis for the beneficial effect of Hes5 in SBMA, we investigated the protein levels of key molecules in SBMA pathogenesis, such as heat shock factor-1 (HSF1), phosphorylated I κ B α and Smad2, in NSC34 cells in which Hes5 is depleted. We found that phosphorylation of Smad2 is substantially down-regulated by siRNA-mediated knockdown of Hes5 despite Smad2 protein levels are not altered (Fig. 8A-C). To strengthen the data, we performed additional experiments. As shown in Fig. 8D, pSmad2 was down-regulated in DHT-treated NSC97Q cells compared with DHT-untreated NSC97Q (Fig. 8D). Moreover Hes5 over-expression induced up-regulation of pSmad2 both in NSC97Q and primary cortical neurons (Fig. 8E, I), indicating that Hes5 protects neurons from the toxic insults of polyglutamine-expanded AR via activation of Smad pathway.

Minor points:

Point 1. There remain quite a few grammatical errors and typos, the authors should review the manuscript for these

Response 1: In response to this comment, our manuscript was proofread by Native speakers of English in Springer Nature Author Services.

Point 2. The control for no over-expression of Dnmt1 in muscle tissue in Fig. 1G,H (and S1-3) is welcome, however the authors suggest localization of Dnmt1 may be most important. Can the authors provide any quantified evidence, such as subcellular localization or 5-methylcytosine levels, in muscle tissue?

Response 2: In response to the comments, we performed quantitative analysis of Dnmt1 and 5mC in muscle. As shown in Supplementary Fig. 1A, the subcellular localization of Dnmt1 is exclusively in the nucleus. Quantitative analysis confirmed that nuclear levels of Dnmt1 are not altered in AR-97Q mouse (Supplementary Fig. 1B). Furthermore, the level of 5mC was also unaffected in the skeletal muscle of AR-97Q mice (Supplementary Fig. 5A and B).

Point 3. Can the authors kindly provide the rationale for choosing RG108?

Response 3: A previous study indicated that RG108 was safe and effective in a mouse model of amyotrophic lateral sclerosis (Chestnut BA et al, J Neurosci 2011). This work was also helpful for us to determine the concentration of the agent. Furthermore our preliminary data demonstrated that RG108 improved the viability of SBMA model cells. These findings made us to choose RG108.

Point 4. Clinical inhibitors for DNMTs, including azacytidine and decitabine, have relatively high toxicity. Was the authors' hope with RG108 to more specifically target the specific DNMT affected? Can the authors provide any evidence of toxicity to blood counts in mice, as is found in patients? If these data are not available, the authors should qualify their statement suggesting DNMT inhibitors may be useful in humans; mice are in a sterile environment and DNMTs are typically administered for short periods of time.

Response 4: Unfortunately, we could not perform blood counts of mice. Therefore, we deleted our description claiming that DNMT inhibitor may be useful in humans (Page 13).

Point 5. Figures 3A-D are missing significance labels

Response 5: We added the significance labels in Fig. 3A-D.

Point 6. Please document the backbone and other characteristics (promoter?) of the Hes5 expression vector and mock vector.

Response 6: According to the comment, we explained the backbone and promoter information of Hes5 expression vector and mock vector in the Methods (Page 19, Line 8-9).

2nd Editorial Decision

6 April 2018

Thank you for the submission of your revised manuscript to EMBO Molecular Medicine. We have now received the enclosed reports from the referees that were asked to re-assess it. As you will see the reviewers are now globally supportive and I am pleased to inform you that we will be able to accept your manuscript pending minor editorial amendments.

Please submit your revised manuscript within two weeks. I look forward to seeing a revised form of your manuscript as soon as possible.

I look forward to reading a new revised version of your manuscript as soon as possible.

***** Reviewer's comments *****

Referee #1 (Comments on Novelty/Model System for Author):

The Authors have improved he models used

Referee #1 (Remarks for Author):

All points were correctly addressed by the Authors

Referee #2 (Remarks for Author):

The authors have done an excellent job of revising the manuscript and have addressed my main concerns. I am not entirely convinced however that their examination of the Smad2 pathway sufficiently distinguished between correlation and causation, and thus request that the authors temper their conclusions regarding Smad by rewriting sentences to include "may" or "possibly", instead of drawing definitive conclusions.

2nd Revision - authors' response

4 May 2018

Referee #1:

All points were correctly addressed by the Authors.

Response: We are truly grateful to you for your review of our work.

Referee #2

The authors have done an excellent job of revising the manuscript and have addressed my main concerns. I am not entirely convinced however that their examination of the Smad2 pathway sufficiently distinguished between correlation and causation, and thus request that the authors temper their conclusions regarding Smad by rewriting sentences to include "may" or "possibly", instead of drawing definitive conclusions.

Response: We are so grateful to you for the constructive evaluation on our revision. We changed the sentence of conclusions using “possibly” according to the comments (Page 2, Line14).

Communication concerning data integrity

17 May 2018

Thank you for sending us the source data for your revised manuscript " DNA methylation inhibitor attenuates polyglutamine-induced neurodegeneration by regulating Hes5" at EMBO Molecular Medicine, and for your patience while we evaluated these additional information. We have carefully checked your figures and source data against one another and have come across several inconsistencies that necessitate further investigation.

As you may know, the journal classifies image aberrations into three levels (<http://embomolmed.embopress.org/classifying-image-aberrations>), and we have classified the issues found in your figures collectively as a serious level I/borderline level II. In line with journal policies, this requires that we involve your research institution to provide an opportunity for quality control and investigation at the institutional level. In our experience, an institutional investigation can help greatly in identifying the causes of the apparent aberrations, as institutions can directly review lab books and interview the authors. It is also in your best interest to clarify the issues in a transparent manner with your employer.

We would kindly invite you to let us know who would be the appropriate colleague in charge of research integrity to be contacted at your research institution.

I look forward to hearing back from you very soon.

RESPONSE FROM INSTITUTION AFTER INSTITUTIONAL INVESTIGATION

In response to your request, we went through the manuscript by Dr. Masahisa Katsuno and checked the figures at the Ad Hoc Committee of the Research Integrity Committee (RIC) of Nagoya University.

As a consequence, we found various mistakes in the figures which should be corrected or reanalyzed. The RIC concluded that the current manuscript does not stand as it is, and recommends the EMBO Molecular Medicine to suspend its publication.

3rd Editorial Decision

12 June 2018

Your manuscript #EMM-2017-08547-V3, entitled "DNA methylation inhibitor attenuates polyglutamine-induced neurodegeneration by regulating Hes5" by Naohide Kondo, Genki Tohnai, Kentaro Sahashi, Madoka Iida, Mayumi Kataoka, Hideaki Nakatsuji, Yutaka Tsutsumi, Atsushi Hashizume, Hiroaki Adachi, Haruki Koike, Keiko Shinjo, Yutaka Kondo, Gen Sobue, and Masahisa Katsuno has been withdrawn following recommendations of the Research Integrity Committee of Nagoya University who conducted a formal investigation.

Should you be able to fix all issues found by the committee, with formal committee agreement that all issues have been fixed, and as long as that the study remains timely and of interest at the time of submission, we would not be opposed to evaluate it once more for publication as if it was an initial submission.

3rd Revision - authors' response

7 January 2019

CONFIRMATION FROM THE INSTITUTE AFTER AUTHORS PERFORMED REVISION

This letter is concerning the manuscript titled 'DNA methylation inhibitor attenuates polyglutamine-induced neurodegeneration by regulating Hes5' y Kondo et al (EMM-2017-08547 V4).

The research Integrity Committee of Nagoya University confirmed that the authors made revisions in response to comments and concerns by RIC. We agree that the journal reevaluates the appropriateness of the revisions.

AUTHOR'S RESPONSE

Ad Hoc Committee of the Research Integrity Committee (RIC)

1. Figure 1. Fig. 1A and the right bottom panel in the source (Dnmt3b)

Authors' statements

They marked lanes 10, 11, and 12 for Dnmt3b in the original and revised source data. They noticed that the correct lanes were 7, 8, and 9, but have not fixed an error.

Conclusion

Authors need to fix an error.

Response 1.

We re-performed western blotting for Dnmt3b and confirmed the lanes which we showed in the main figure correspond to the lanes we marked in the revised source file of Fig. 1A. We also revised the quantitative analysis (Fig. 1B) due to the redo of western blot.

2. Figure 1. Fig. 1G and the right middle panel in the source (GAPDH)**Authors' statements**

The excised bands for GAPDH in the original figure were wrong. In the revision, they claimed that the dense upper bands were correct GAPDH, and they revised Figure 1G and a frame in the source data accordingly.

Concern

The predicted molecular weight of GAPDH = 36 kDa. The original bands look correct and the revision looks wrong.

Conclusion

The authors need to recognize which bands are indeed GAPDH.

Response 2.

As we mentioned in our response 3 and 4, we re-performed western blotting for Dnmt1 using our original mouse skeletal muscle samples and run the same samples on a separate gel for GAPDH. We confirmed that GAPDH appears just below the molecular weight marker of 37 kDa. We revised Fig. 1G, and marked the right bands in the revised source file of Fig. 1G.

3. Figure 1. Top two panels in the source for Fig. 1 (Dnmt1)**Concern**

Mobilities of Dnmt1 are different between these two panels.

Conclusion

Additional analysis is required. The authors need to recognize which bands are indeed Dnmt1 by performing Western blotting of cells that are knocked down for Dnmt1 and/or overexpress Dnmt1. Alternatively, the authors may immunoprecipitate Dnmt1 with anti-Dnmt1 antibody #1, and immunoblot the precipitated sample with anti-Dnmt1 antibody #2.

Response 3.

We noticed that the different results stemmed from the fact that we used antibodies with different Lot numbers. To solve this problem, we tested 3 kinds of antibodies (ab188453, Abcam; ab13537, Abcam; and sc20701, Santa Cruz), and found that ab188453 (1:1000) has the best ability to detect the bands. Using this antibody, we confirmed that Dnmt1 appears between 150 kDa and 250 kDa markers in the western blotting with the sample of NSC97Q cells treated with anti-Dnmt1 siRNA as a negative control (please refer the revised source file for Fig. 1A_Dnmt1). Based on the results of the additional western blotting, we changed the bands of Dnmt1 in Fig. 1A and 1G, and re-performed the quantitative analysis for Fig. 1B and 1H.

4. Figure 1. Right top and middle panels in the source for Fig. 1 (GAPDH)**Concern**

They first performed Western blotting (WB) with anti-Dnmt1 antibody (right top panel). Without stripping off the anti-Dnmt1 antibody, they re-probed the blot with anti-GAPDH antibody (right middle panel). Dnmt1 WB had many nonspecific bands, and some bands overlapped with GAPDH bands. They cannot estimate the amount of GAPDH without stripping off anti-Dnmt1 antibody.

Conclusion

Reanalysis is required. Strip off anti-Dnmt1 antibody or run samples on a separate gel for quantification of GAPDH.

Response 4.

As we mentioned in our Response 2, we re-performed western blotting for Dnmt1 with our original skeletal muscle samples of mice, and run the same samples on another gel for GAPDH to avoid the effects of stripping on re-probing. Based on the results of the additional western blotting, we changed the bands of GAPDH in Fig. 1A and 1G.

5. Figure 2. Fig. 2A and the left 2nd panel in the source for Fig. 2AC (Dnmt3a)**Concern**

The molecular weight of Dnmt3a is more than 100 kDa, but looks less than 100 kDa. The RIC assumes that there was an error in the size marker.

Conclusion

Confirm the size markers and revise the source panel.

Response 5.

We re-performed western blotting using our original samples and confirmed that the bands of Dnmt3a appear above the marker of 100 kDa. Based on the results of the additional western blotting, we changed the bands of Dnmt3a and GAPDH in Fig. 2A, and re-performed the quantitative analysis.

6. Figure 2. Fig. 2C and the left 3rd panel in the source for Fig. 2AC (Dnmt1)

Authors' statements

Lanes 2-5, 6-9, and 10-13 were from triplicated experiments, and each group of bands looked similar. Dnmt1 in Fig. 2C were excised from lanes 4 and 5, but we inadvertently marked lanes 8 and 9 in the original source data.

Conclusion

Authors made an appropriate correction in the revised source data.

Response 6.

To clarify the alteration of Dnmt1 protein levels we re-performed anti-Dnmt1 western blotting using our original samples, and changed the bands of Dnmt1 in Fig. 2C and quantitative data.

7. Figure 4. Fig. 4C and the left top panel in the source for Fig. 4 (GAPDH)

Authors' statements

They could not find the source data for GAPDH in the original Fig. 4C. They thus used GAPDH bands of the same blot with different exposure in the source data. They excised GAPDH bands from the source data, and revised Fig. 4C. The first author looked for the original source data from Toronto, but the limited access to the source data disabled identification of the source data. After he came back to his laboratory, he has found the source data.

Conclusion

Authors need to make sure that GAPDH bands in Fig. 4C and in the revised source data indeed represent correct samples.

Response 7.

The original data was lost while the first author moved from our lab to Toronto. We thus re-performed anti-AR western blotting using our original samples, and run the same samples for GAPDH. Based on the results of the additional western blotting, we changed the bands of AR and GAPDH in Fig. 4C.

8. Figure 4. Fig. 4C and the left bottom panel in the source for Fig. 4 (GAPDH)

Concern

The mark for 37 kDa does not match the position of a ladder.

Conclusion

Authors need to confirm the position of size markers.

Response 8.

As shown in the revised source file of Fig. 4C, we confirmed that GAPDH appears just below the molecular weight maker of 37 kDa.

9. Figure 4. Fig. 4C and the two left panels in the source for Fig. 4 (Dnmt1 and GAPDH)

Authors' statements

Lane marks for Dnmt1 and GAPDH were wrong in the original and revised source data. Triplicated experiments were run on lanes 3-4, 5-6, and 7-8. Lanes 3-4 should have been marked, but lanes 2-3 were erroneously marked.

Conclusion

Authors need to correct an error.

Response 9.

As we mentioned in our response 7, we re-performed anti-AR western blotting, run the same samples on a separate gel for GAPDH (revised Fig. 4C), and re-performed the quantitative analysis (revised Fig. 4D).

10. Figure 4. Fig. 4C and the top two panels in the source for Fig. 4 (Dnmt1 and Chat) Concern

For both panels, triplicated experiments were run on lanes 3-4, 5-6, and 7-8. RG108 appears to similarly increase AR aggregates and Chat in lanes 4 and 8. Similarly, RG108 appears to similarly decrease AR aggregates and Chat lane 6. However, authors conclude that RG108 had no effect on AR aggregation (Fig. 4CD), but significantly increased Chat expression (Fig 4HI). In addition, AR aggregates in Fig. 4C do not appear to identical to the source data in the left upper panel.

Conclusion

Scrutinized reanalysis is required. The source data do not appear to support the authors' conclusions.

Response 10.

As explained in our response 7, we re-performed anti-AR western blotting, and confirmed that RG108 did not reduce the high molecular weight complex of AR (revised Fig. 4C). We also re-performed anti-Chat western blotting, and run the same samples for GAPDH. Based on the results of these additional experiments, we changed the bands of Chat and GAPDH (revised Fig. 4H), and re-performed the quantitative analysis (revised Fig. 4I).

11. Figure 5. Fig. 5A and the right bottom panel in the source for Fig. 5AB (GAPDH) Authors' statements

In the original Fig. 5A, GAPDH bands were erroneously excised from non-specific bands. Correct GAPDH bands were excised in the revision.

Conclusion

The error was appropriately corrected in the revision.

Response 11.

To confirm Dnmt1 protein levels using the Dnmt1 antibody (ab188453) we re-performed western blotting using our original samples, and run the same samples for GAPDH to prevent that unstripped bands affect the detection of GAPDH. Based on the results of the additional western blotting, we changed the bands of Dnmt1 and GAPDH (revised Fig. 5A), and re-performed the quantitative analysis (revised Fig. 5B).

12. Figure 7. Fig. 7G and the two right panels in the source for Fig. 7EG (AR monomer and GAPDH) Authors' statements**Authors' statements**

The source data for AR monomer and GAPDH used in the original Fig. 7G could not be identified in an archive of electronic media. Authors instead found an image with longer exposure time of the same blot. In the revised Fig. 7G, authors excised AR monomer bands and GAPDH bands from the source data with longer exposure time in the revision.

Conclusion

The author should have archived the source data that they used in their manuscript. However, authors made an acceptable revision.

Response 12.

None (revision had been made).

13. Figure 7. Fig. 7EG and the bottom two panels in the source for Fig. 7EG (GAPDH) Concern**Concern**

The molecular weight of GAPDH is 36 kDa. A marker for 37 kDa appears to be wrongly marked.

Conclusion

Authors need to scrutinize size markers throughout their source data.

Response 13.

As we mentioned in our response 12, we re-performed anti-GAPDH western blotting, and confirmed that GAPDH appears just below the molecular weight maker of 37 kDa.

14. Figure 7. Figure 7EG and the left top panel in the source for Fig. 7EG (Hes5)**Authors' statements**

Authors were not aware of the difference between the predicted molecular weight (18 kDa) and the molecular weight of the stained band for Hes5 (~35 kDa).

Concern

The molecular weight of Hes5 is 18 kDa. The Hes5 bands marked by authors in Figs. 7EG, 8I, and S15BE were ~35 kDa and were much higher than 18 kDa. Datasheets of anti-Hes5 antibody by Novusbio and SantaCruz (M-104) show that Hes5 is stained at ~35 kDa, but the validity of these antibodies need to be scrutinized (Authors used the SantaCruz antibody). In addition, the position of the 50-kDa marker looks incorrect in the source data for Fig. 8I.

Conclusion

Additional analyses are required. Authors need to recognize the identity of Hes5 by performing Western blotting of cells that are knocked down for Hes5 and/or overexpress Hes5. Knockdown of Hes5 was already shown in S15B, but intensities of multiple bands were reduced. Confirmation by overexpression experiments is likely to be required.

Response 14.

We clarified that the height of HA-tagged Hes5 is between 20 and 25 kDa in anti-HA western blotting (attached below). We also found that the anti-Hes5 antibody we used in our original experiments (sc13859, Santa Cruz) detects bands between 20 and 25 kDa. Therefore, we re-trimmed such bands in Fig. 8C, 8E, 8F, and 8I, and re-performed the quantitative analysis of them. However, as our original results did not clearly detect Hes5 in Fig7B, 7E, S15B, and S15E, we re-performed additional western blotting of the original samples by shortening the electrophoresis time. According to the Committee's comments, we run the same samples for western blotting of GAPDH to avoid the effects of stripping. Based on the results of these additional experiments, we changed the bands of Hes5 and GAPDH in Fig. 7B, 7E, S15B, and S15E, and re-performed the quantitative analysis of them. We confirmed that over-expression and knock-down of Hes5 were successfully done in each experiment.

**15. Figure S7. Fig. S7B and the left bottom panel in the source for Fig. S7 (GAPDH)****Concern**

The molecular weight of GAPDH is 36 kDa. A marker for 37 kDa appears to be wrongly marked.

Conclusion

Authors need to scrutinize size markers throughout their source data.

Response 15.

We re-analyzed the original blot, and confirmed that GAPDH appears just below the molecular weight maker of 37 kDa (revised source file of Fig S7B).

16. Figure S7. Fig. S7AC (GAPDH)

Concern

Source data for GAPDH of Fig. S7AC are not indicated.

Conclusion

Show the relevant source data.

Response 16.

We re-performed anti-Dnmt1 western blotting using our original samples, and run the same samples for GAPDH. Based on the results of the additional western blotting, we changed the bands of Dnmt1 and GAPDH in Fig. S7A, and re-performed the quantitative analysis. We also add the result of anti-GAPDH re-probing of the blot shown in Fig. S7C, which was not included in our previous source file.

17. Figure S7. Fig. S7B, and the right upper and left bottom panels in the source for Fig. S7 (Dnmt3a and GAPDH)

Authors' statements

They changed the lane marks on the revised source data, but they claim that the original lane marks were correct.

Conclusion

Correct the lane marks again.

Response 17.

We re-marked the right bands in the source file of Fig. S7B.

18. Figure S7. Fig. S7AC, and the upper two and right bottom panels in the source for Fig. S7 (Dnmt1 and Dnmt3a and Dnmt3b)

Authors' statements

The authors marked lanes 5 and 6 in the original source data, but they should have been lanes 3 and 4. The revised source data were marked correctly. They noticed that the lane marks on the right bottom panel for Dnmt3b should be Fig. S7C, but were erroneously marked as Fig. S7B.

Conclusion

Correct lanes were marked in the revision. Correct the tag from Fig. S7B to Fig. S7C.

Response 18.

As we mentioned in our responses 16 and 17, we re-performed western blotting for Fig. S7A, and re-marked the right bands in the source file of Fig. S7B and C, so that the source file corresponds to the Figures. We also corrected the label of Fig. S7C, which had been erroneously shown as "Fig. S7B".

19. Figure S7. Fig. S7AB and the left top panel in the source for Fig. S7 (Dnmt1)

Concern

The authors marked the strongest bands as Dnmt1 in the original and revised source data. However, knockdown of Dnmt1 did not reduce the signal intensity, although the authors showed that the signal intensities were reduced to about a half of controls. Dnmt1 marked in the source data for Figs. 1A, 2A, and 5A was slightly above a 150-kDa marker, whereas Dnmt1 marked in the source data for Fig. 1G was ~250-kDa. In Fig. S7A, the marked Dnmt1 is far above a 150-kDa marker. The RIC can recognize faint bands above the 150-kDa marker and below the strongest bands. Signal intensities of these faint bands were reduced by knockdown of Dnmt1.

Conclusion

Scrutinized reanalysis is required for Figs. 1A, 1G, 2A, 5A, and S7AB for identity of Dnmt1. This is repetition of concern #3. The authors need to identify Dnmt1 by performing Western blotting of cells that are knocked down for Dnmt1 and/or overexpress Dnmt1.

Response 19.

As we mentioned in our responses 3, 6, and 7, we re-performed anti-Dnmt1 western blotting, run the same samples for GAPDH, and re-performed the quantitative analysis for Fig. 1A, 1G, 2C, and S7A. We did so for Fig. 2A, 5A, 5C, and S8A as well.

20. Figure S15. Figure S15B and the left top panel in the source for Fig. S15**Authors' statements**

Molecular weight markers were erroneously marked in all four panels in the original source data. Authors did not scrutinize paring pictures that showed positions of the markers. The markers were corrected in the revision.

Conclusion

Erroneous marking of molecular weights was appropriately corrected in the revision.

Response 20.

We re-performed anti-Hes5 western blotting, and confirmed the height of the bands of Hes5 and size markers (revised Fig. S15B and E).

21. Figure S15. Figure S15BE and top two panels in the source for Fig. S15 (Hes5)**Authors' statements**

Bands for Hes5 (S15BE) and Gapdh (S15B) were erroneously marked in the original source data, and were corrected in the revision.

Concern

The concern is identical to concern #13. The indicated molecular weight of Hes5 (~35 kDa) is much different from it predicted molecular weight of 18 kDa.

Conclusion

Additional analyses are required. Erroneous marking of bands for Hes5 and Gapdh were corrected in the revision, but identity of Hes5 should be revealed.

Response 21.

As we mentioned in our response 20, we re-performed anti-Hes5 western blotting, and confirmed that the height of the bands of Hes5 is between 20 and 25 kDa (revised source file of Fig. S15B and E).

22. Results for Figure 6FG (Hes5 expression after RG108 treatment)**Concern**

In a section for "RG108 recovers Hes5 expression" in Results, authors state "it (Hes5) was diminished by RG108 treatment (Fig. 6F and G)". This should be "enhanced" or "upregulated".

Conclusion

The error should be corrected.

Response 22.

We corrected "diminished" to "up-regulated" in the text (Page 10, Line 13).

4th Editorial Decision

11 January 2019

Thank you for submitting your newly revised manuscript to EMBO Molecular Medicine. We have now carefully evaluated the figures along with the source data provided as well as the reply to the RIC letter that you kindly provided.

While in principle, we are supportive of publication, still a few items remain to be corrected and some have to be added as follows:

- 1) Figures vs. source data: please fix!
 - the bands in the figure do not seem to be the same as in the source data in
 - Figure 5C_Dnmt3b
 - Figure 8A_Smad2
 - Figure 8D_Smad2
 - Figure 8F_Smad2
 - Figure 8I_GAPDH

Appendix Figure S12_SH97Q (different exposures in figure that contradicts the appendix figure) both M and U

Appendix FigureS13C_NSC97Q

- Labelling mistake we believe in Appendix Figure S3B_GAPDH (labelled twice as such, for cerebellum), we believe that one of them matches the cortex samples

I look forward to receiving a new revised version of your manuscript as soon as possible.

4th Revision - authors' response

1 March 2019

We are most grateful to you for your supportive comments on our manuscript. We revised Figures, Source data and manuscript according to the comment 1) 2) 5) and 6), and added created Author checklist, The Paper Explained, Synopsis, and Visual abstract in response to the comments 3) 4) 7) 8) and 9). Detailed responses are shown below.

1) Figures vs. source data: please fix!

- the bands in the figure do not seem to be the same as in the source data in

Figure 5C_Dnmt3b

Figure 8A_Smad2

Figure 8D_Smad2

Figure 8F_Smad2

Figure 8I_GAPDH

Appendix Figure S12_SH97Q (different exposures in figure that contradicts the appendix figure) both M and U

Appendix FigureS13C_NSC97Q

Response: In accordance to your instruction, we revised the Figures so that they match the source data, by removing any manipulation of angle and contrast of the images.

- Labelling mistake we believe in Appendix Figure S3B_GAPDH (labelled twice as such, for cerebellum), we believe that one of them matches the cortex samples

Response: As you pointed out we had a mistake to label the data. We changed the label in the source of Appendix Figure S3B_GAPDH from “cerebellum” to “cortex”.

Corresponding Author Name: Masahisa Katsuno

Journal Submitted to: EMBO molecular medicine

Manuscript Number: EMM-2017-08547